# Electrocatalytic hydrogenation of quinolines with water over a fluorine-modified cobalt catalyst

Shuoshuo Guo [1,4], Yongmeng Wu [1,4], Changhong Wang [1], Ying Gao [1], Mengyang Li [1,2], Bin Zhang [1,3] ✉ & Cuibo Liu [1,3] ✉

Room temperature and selective hydrogenation of quinolines to 1,2,3,4-tetrahydroquinolines using a safe and clean hydrogen donor catalyzed by cost-effective materials is significant yet challenging because of the difficult activation of quinolines and $H_2$. Here, a fluorine-modified cobalt catalyst is synthesized via electroreduction of a Co(OH)F precursor that exhibits high activity for electrocatalytic hydrogenation of quinolines by using $H_2O$ as the hydrogen source to produce 1,2,3,4-tetrahydroquinolines with up to 99% selectivity and 94% isolated yield under ambient conditions. Fluorine surface-sites are shown to enhance the adsorption of quinolines and promote water activation to produce active atomic hydrogen (H*) by forming $F^-$-$K^+(H_2O)_7$ networks. A 1,4/2,3-addition pathway involving H* is proposed through combining experimental and theoretical results. Wide substrate scopes, scalable synthesis of bioactive precursors, facile preparation of deuterated analogues, and the paired synthesis of 1,2,3,4-tetrahydroquinoline and industrially important adiponitrile at a low voltage highlight the promising applications of this methodology.

1,2,3,4-Tetrahydroquinoline (THQ) skeletons are prevailing and important structural motifs that commonly reside in bioactive entities and natural products[1,2]. At present, hydrogenation of quinolines catalyzed by transition metals using molecular hydrogen ($H_2$) as the hydrogen source at relatively high reaction temperatures (usually ≥100 °C) is still a prevailing method to synthesize 1,2,3,4-tetrahydroquinolines, which has been a long-standing subject in synthetic chemistry and serves as a model reaction to evaluate the performance of a newly developed catalyst or hydrogenation system[3–13]. For example, Corma and coworkers made an important advance in realizing highly chemo- and regioselective hydrogenation of quinoline derivatives over a nanolayered cobalt-molybdenum-sulfide (Co-Mo-S) catalyst at 110–150 °C under 12 bar of $H_2$[14]. The Beller group fabricated the heterogeneous N-doped carbon-modified iron-based catalysts for the selective hydrogenation of quinolines and (iso)quinolones by using

40–50 bar of $H_2$ at 130–150 °C[15]. Despite these impressive achievements, the storage and transportation of flammable $H_2$ often require special apparatuses, cautious operations, and additional manpower input. Additionally, hydrogenation of the benzene ring usually occurs under thermocatalytic conditions, leading to the generation of 5,6,7,8-tetrahydroquinoline and decahydroquinoline byproducts. Therefore, searching for an efficient and sustainable method for the selective hydrogenation of quinolines to THQs by applying a green and easy-to-handle hydrogen donor (e.g., $H_2O$) at room temperature (RT) is highly significant. Such a technique will achieve distributed manufacturing of quinoline hydrogenation where $H_2$ as a hydrogen source is not readily available, thus complementing the state-of-the-art thermocatalytic hydrogenation of quinolines by using $H_2$.

Recently, electrochemistry has become increasingly significant in the synthesis field because of its mild, efficient, and environmentally

[1]Department of Chemistry, School of Science, Institute of Molecular Plus, Tianjin University, 300072 Tianjin, China. [2]Green Catalysis Center, College of Chemistry, Zhengzhou University, 450000 Zhengzhou, China. [3]Haihe Laboratory of Sustainable Chemical Transformations, 300192 Tianjin, China. [4]These authors contributed equally: Shuoshuo Guo, Yongmeng Wu. ✉e-mail: bzhang@tju.edu.cn; cbliu@tju.edu.cn

benign properties[16–22]. Electrocatalytic hydrogenation powered by renewable electricity has gradually been proven to be an attractive and promising approach to obtaining value-added hydrogenated products by the direct utilization of clean and safe $H_2O$ as the hydrogen donor[23–25]. Importantly, deuterated chemicals with improved biological or physicochemical properties compared with their hydrogenated analogs due to the kinetic isotope effect of deuterium (D) can be economically and expediently synthesized by employing inexpensive and safe $D_2O$[26,27]. Current studies are predominantly focusing on the hydrogenation of $CO_2$, nitrate[28–31], and other easily reducible organic substrates (e.g., nitros, aldehydes, ketones, alkynes, halides, and nitriles)[23–27]. However, the electrocatalytic hydrogenation of quinolines with high resonance stability still faces a technological challenge. The Lei group reported a well-designed electrochemical arylation of electron-deficient arenes through reductive activation using a Pt plate cathode. However, the yields of coupling products of quinoline derivatives are low[32]. We speculate that these inferior performances may be ascribed to the lack of suitable materials to effectively activate quinolines. Given the importance of THQs and related compounds, it is highly desirable to synthesize an advanced material to efficiently promote the activation of quinolines and water, which will be conducive to boosting the activity and selectivity of quinoline hydrogenation.

Electrode materials play a crucial role in electrochemical transformation reactions[33,34]. Co materials are documented to be efficient catalysts for the hydrogenation of quinolines and other *N*-heterocycles in traditional thermocatalysis[12,14,35–37]. In addition, Co-based cathodes are widely studied in electrochemical water splitting and nitrate electroreduction due to their low cost, easy preparation, and high stability[38–40]. Generally, regulating the electron structure or coordination environment of a metal electrocatalyst can modulate the adsorption behaviors of the reactant, intermediate, and product on the catalyst surface and the affinity of the active hydrogen atom (H*) from $H_2O$ dissociation, thus influencing the reaction pathway or efficiency. Halogen modification has been reported as an efficient route to tune the electron distributions of the catalytic center, thus enhancing the activity and selectivity toward the $CO_2$ reduction reaction ($CO_2RR$) or promoting the formation of H* from $H_2O$ dissociation[41,42]. Furthermore, due to the electron-donating property of the saturated *N*-heterocycle in THQ, the benzene ring is more likely to lose rather than obtain an electron before initiating subsequent reactions under electro- or photocatalytic reaction conditions[43]. It is harder to further hydrogenate under our electroreduction conditions, thus improving the regioselectivity of the electrocatalytic hydrogenation of quinolines. In this regard, the introduction of halogen to nanostructured Co is supposed to enable efficient electrocatalytic quinoline hydrogenation to THQs with high selectivity.

Here, we propose and validate that the highly electronegative fluorine (F) on the cobalt (Co) surface can promote the adsorption of quinolines and the activation of $H_2O$. Thus, a nickel foam (NF)-supported F-modified Co electrocatalyst (denoted as Co-F) is synthesized via electroreduction of the Co(OH)F nanowire (NW) precursor, which demonstrates high activity and selectivity for the electrocatalytic hydrogenation of a series of quinolines and other *N*-heterocycles with water at room temperature. The gram-scale synthesis of pharmacologically related precursors, facile preparation of D-containing THQs, and paired production of value-added products make this electrocatalytic strategy stand out from the current approaches.

## Results

### Calculation prediction for designing a Co-F electrocatalyst
Metal catalytic centers with a low electron density can usually enhance the adsorption of substrates and reaction intermediates, thus improving catalytic performance. Therefore, we selected F as a modifier to investigate its effects on the electrocatalytic performance of Co because of its higher electronegativity and smaller atomic radius. Next, density functional theory (DFT) calculations are performed to provide further support for the rational design of an effective Co electrocatalyst. After optimizing the most stable adsorption modes (Supplementary Fig. 1), quinoline **1a** prioritizes flat adsorption on the Co(111) surface. Figure 1a reveals that the presence of surface F can significantly enhance the adsorption of **1a**, which will facilitate electron transfer between **1a** and the electrode. Additionally, the much lower adsorption energies ($E_{ads}$) of **2a** than that of **1a** imply that the product is easier to desorb from the Co cathode (Fig. 1b). Furthermore, the Gibbs free energy for H* formation ($\Delta G_{H*}$) on Co-F is more negative than that on pure Co (−0.46 vs. −0.38 eV, Fig. 1c). This indicates that the Co-F cathode is favorable for generating H* via $H_2O$ dissociation, which will benefit the hydrogenation of **1a**. These considerations and theoretical results encourage us to synthesize a Co catalyst with surface F, which will serve as an efficient electrode for achieving the electrocatalytic hydrogenation of quinolines to THQs with high reaction efficiency and regioselectivity via water electroreduction.

### Synthesis and characterization of a Co-F electrocatalyst
Electroreduction of precursors offers an important addition to the toolbox of chemical conversion synthesis of highly active materials[41]. Halide anions in transition metal halides might leach and adsorb on the reconstructed surface under a given potential and a pH value. Additionally, binder-free self-supported three-dimensional (3D) electrodes can allow more active sites to be exposed and improve electrical conductivity. Here, a Co(OH)F NW precursor was synthesized on a NF substrate through a solution-chemical process (Supplementary Note 1). Scanning electron microscopy (SEM) images, transmission electron microscopy (TEM) images, energy-dispersive X-ray spectroscopy (EDS) elemental mapping, X-ray diffraction (XRD) patterns, and X-ray photoelectron spectroscopy (XPS) spectra suggest the successful preparation of Co(OH)F NW (Supplementary Figs. 2 and 5a, b, and Supplementary Note 2). Then, a facile electroreduction treatment of Co(OH)F in 1.0 M KOH at −1.2 V vs. Hg/HgO (potentials in this work are all referred to as Hg/HgO unless otherwise stated) was used to synthesize the NF-supported surface-F-modified Co catalyst. The chronoamperometry *i-t* curve and in situ Raman spectra (Supplementary Fig. 3) record the transformation process of Co(OH)F. In the Co 2*p* XPS spectra (Fig. 1d), the appearance of peaks at ~778.0 and 793.1 eV belong to $Co^0$ $2p_{3/2}$ and $Co^0$ $2p_{1/2}$, confirming the formation of metallic Co(0)[40]. The deconvoluted peaks located at 779.3, 781.3, 795.3, and 797.1 eV are assigned to oxidized Co due to the inevitable oxidation of the Co surface during the XPS test, which shows negative shifts compared with that of Co(OH)F[40]. Moreover, the peak at 684.05 eV in the F 1*s* spectrum is ascribed to the $F^-$ ion, which displays the same valence state in both samples (Fig. 1e). However, the F 1*s* peak intensity in the reduced sample is much lower, supporting the EDS result. In situ X-ray absorption spectroscopy (XAS) was applied to study the real structural evolution of Co(OH)F under electrochemical conditions. The Co *K*-edge X-ray absorption near-edge structure (XANES) spectrum of the reduced sample at −1.2 V exhibits similar features to that of the reference Co foil, demonstrating that the reduced sample mainly consists of metallic Co. However, the Co absorption edge position of the reduced sample is located between the Co foil and Co(OH)F, suggesting a higher valence state of Co-F than Co foil (Fig. 1f and its insert). This may be ascribed to the existence of low-coordination Co caused by $F^-$ stripping during the electroreduction process and a small part of unreduced Co(OH)2. In addition, quantitative calculation of reduced sample was performed by the linear combination, demonstrating an ~65% fraction of metallic Co with 35% remaining Co(OH)2. The Fourier transform extended X-ray absorption fine structure (EXAFS) spectrum shows one main peak at 2.1 Å,

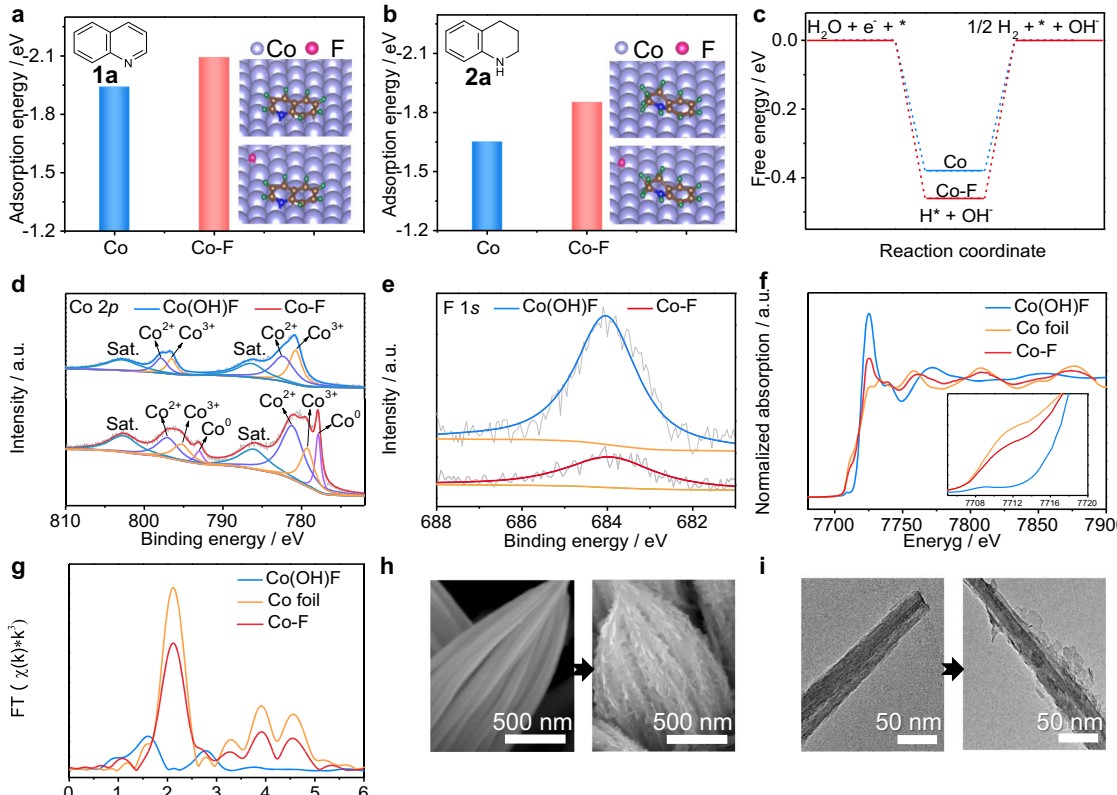

**Fig. 1 | Synthesis of Co-F NWs by electroreduction of Co(OH)F NWs.**
**a**, **b** Comparisons of $E_{ads}$ of **1a** and **2a** on pure Co and Co-F, respectively (insert: the stable adsorption modes of **1a** and **2a**). **c** The calculated $\Delta G_{H^*}$ over pure Co and Co-F. **d** Co 2$p$ XPS spectra and **e** F 1$s$ XPS spectra of the Co(OH)F and Co-F catalysts. **f** In situ Co K-edge XANES spectra and **g** EXAFS spectra of Co(OH)F NW at −1.2 V vs. Hg/HgO in 1.0 M KOH (the spectra of Co foil and Co(OH)F are also shown for reference.). **h** SEM and **i** TEM images of Co(OH)F and Co-F catalysts.

corresponding to the first coordination shell of Co–Co (Fig. 1g). The peak at around 1.6 Å may belong to Co–O path caused by the oxidation of metallic Co by dissolved oxygen or Co(OH)$_2$. According to the EXAFS data fitting results (Supplementary Fig. 4), the average coordination number of Co of the reduced sample is ca. 6.5, which is lower than that of Co foil, and the mean bond length is 2.49 Å (Supplementary Table 1). SEM images (Fig. 1h and Supplementary Fig. 6a, b) reveal the retained nanowire structure after electroreduction. However, the surface becomes rough. TEM images show that the reduced sample is composed of low crystalline nanosheets and crystalline regions with a distinct boundary between them (Fig. 1i and Supplementary Fig. 6c, d). The low crystalline structure should belong to the formed metallic Co. The crystal lattice fringe with an interplanar distance of 0.278 nm is attributed to the (100) plane of Co(OH)$_2$, as validated by the XRD pattern (Supplementary Fig. 5a). However, Co(OH)$_2$ is difficult to transform into a pure Co phase, even prolonging the reaction time or applying more negative potentials. The as-formed low crystalline layer of Co inhibits further reduction of inner Co(OH)$_2$. The uneven distribution of Co in the EDS elemental mapping images further verifies that Co(OH)$_2$ is surrounded by Co (Supplementary Fig. 6e). Supplementary Fig. 5b, c show a much lower F content of the reduced sample than that of the precursor (6.08 vs. 33.09%), implying the severe leaching of F during the electroreduction process. The ex situ and in situ characterizations indicate that the Co(OH)F NW can be electroreduced to surface-F-modified low-coordinated Co with a small fraction of internal Co(OH)$_2$ (to simplify, Co-F NW is used for the reduced sample), which will work as an electrocatalyst for the hydrogenation of quinolines with H$_2$O.

## Electrocatalytic hydrogenation of quinolines with H$_2$O over the Co-F cathode

After obtaining the Co-F NW, it is quickly used as the cathode for the electrocatalytic hydrogenation of quinolines. The reaction was carried out in a divided three-electrode system by using 0.1 mmol of **1a** as the model substrate with a mixed solution of 1.0 M KOH and dioxane (6:1 v/v, 7 mL) (for the reaction setup, see Supplementary Fig. 7a). No clear differences are observed from the linear sweep voltammetry (LSV) curves before and after adding **1a** into the electrolyte (Supplementary Fig. 8a). We speculate that the reduction peak of **1a** may be overlapped by the electrolysis of water over Co-F NW. Potential-dependent electrochemical experiments were carried out to screen the optimal potential for 8 h. Figure 2a shows that **1a** can be electroreduced from −1.0 to −1.3 V. At the optimal potential of −1.1 V (Fig. 2b), nearly full conversion of **1a** is finished in 6 h to produce **2a** with 99% selectivity (Supplementary Fig. 9, and Supplementary Notes 3 and 4). This result demonstrates that 6 h is enough for the electrocatalytic hydrogenation of **1a** at −1.10 V vs. Hg/HgO. Thus, 6 h was selected for subsequent experiments. No byproducts related to the hydrogenation of the phenyl ring are observed at any test potential (Fig. 2a), and the selectivity of **2a** does not decline even at a negative potential and over a long period (Supplementary Fig. 8b), revealing the excellent regioselectivity. In addition, our Co-F cathode demonstrates the highest conversion of **1a** among all the tested Co-based materials. The other electrodes, including Pt, Pd, and Ni$_2$P, display high hydrogen evolution reaction (HER) activities, and Cu and glass carbon (GC) show poor activities for the HER, but all of them are inferior to Co-F for the hydrogenation of **1a** under the same reaction conditions (Fig. 2c). The NF support shows almost no activity toward **1a** electroreduction even

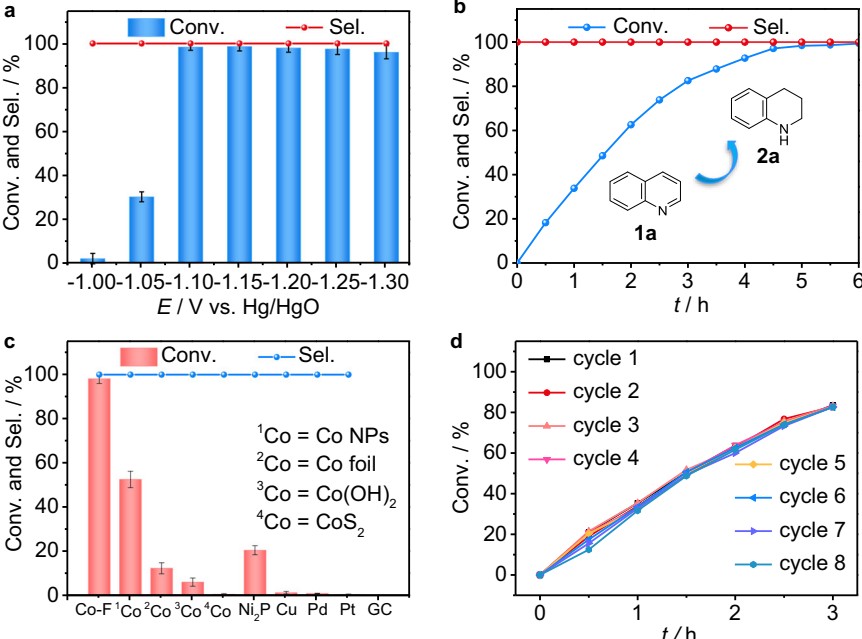

**Fig. 2 | Performance of the electrocatalytic hydrogenation of quinolines over a Co-F cathode. a** Potential-dependent and **b** time-dependent **1a** conversion (Conv.) and **2a** selectivity (Sel.) over the Co-F. **c** Comparison results of converting **1a** to **2a** over other electrodes. **d** Cycle-dependent **1a** Conv. over the Co-F within 3 h plotting by conversion as a function of time for each cycle. Reaction conditions: **1a** (0.1 mmol, 14.28 mmol L$^{-1}$), a mixed solution of 1.0 M KOH/dioxane (6:1 v/v, 7 mL), RT. **a** 8 h; **b, c** −1.1 V, 6 h; **d** −1.1 V, 3 h. Error bars correspond to the standard deviation of three independent measurements.

at different applied potentials (Supplementary Fig. 10). Furthermore, we observe an obvious induction period within the initial 0.5 h when using the Co(OH)F precursor as the cathode under similar reaction conditions (Supplementary Fig. 11). This may further demonstrate that the electrochemical activation of Co species is crucial for this hydrogenation reaction. Moreover, to evaluate the recyclability of Co-F, this electrode is repeatedly used for the next electrochemical experiments by adding the same amount of **1a** after being washed several times with ethanol and deionized water. Time-dependent transformations reveal that the conversion of **1a** gradually decreases from the sixth cycle within the initial 0.5 h (Fig. 2d). However, it can still approach 83% conversion within 3 h, which is the same as that of the former five cycles. Additionally, we also tested the performance stability of Co-F in our screened optimal 6 h (Supplementary Fig. 12). There was no apparent decline in the conversion of **1a** and selectivity of **2a** in 7 runs. These results may demonstrate the relatively stable performance of Co-F for the electrocatalytic hydrogenation of **1a**. However, for the eighth and ninth runs, the decreased conversion of **1a** may be mainly ascribed to the loss of surface F (Supplementary Fig. 13) and the deactivation of Co catalytic sites caused by the nitrogen moiety of the substrate or product.

## Mechanistic studies

The high-performance origin of the electrocatalytic hydrogenation of **1a** over Co-F is further studied. First, compared with Co(OH)F, Co-F exhibits a higher double-layer capacitance ($C_{dl}$) and a smaller charge-transfer resistance $R_{CT}$ (Supplementary Fig. 14), suggesting a larger electrochemical active surface area (ECSA) and fast reaction kinetics. These factors will provide more active sites and accelerate electron transfer between the electrode and **1a**/H$_2$O, thus enhancing the electrocatalytic performance. Second, a deuterium-labeling experiment verifies H$_2$O serving as the hydrogen source (Supplementary Fig. 15). Third, we observe a lower onset potential and a significantly enhanced Conv. of **1a** after adding NaF into the electrolyte over a Co foil cathode (Fig. 3a, b). These results suggest that F$^-$ plays a promotional role in both HER and **1a** hydrogenation, which may be due to the enhanced

adsorption of **1a** and activation of H$_2$O by F$^-$ (Fig. 1a-c). However, the Conv. of **1a** is still much lower over the Co foil cathode after the extra addition of NaF than that of Co-F (67.1 vs. 98.2%) within the same reaction time at −1.2 V. The enhanced performance of Co-F can be ascribed to the nanostructured morphology, low crystalline surface, and an appropriate amount of F$^-$ adsorbent.

Additionally, surface-adsorbed anions (e.g., S$^{\delta-}$, F$^-$) have been reported to play a vital role in accelerating H$_2$O activation to form active H* species in alkaline media[41,44–46]. An anion-hydrated cation network (X$^{\delta-}$-K$^+$(H$_2$O)$_n$, where $n$ refers to the number of ionic hydrations) will be formed in the Helmholtz layer structure via non-covalent Coulomb interactions[47,48], contributing to the dissociation of H$_2$O to form active H* species, which is usually a slow step in alkaline solution. Control experiments were performed to validate whether a similar promoting effect was involved in our reaction (Fig. 3c, d). When we apply 1.0 M tetramethylammonium hydroxide (TMAH) solution as the electrolyte, inferior performances of both HER and **1a** Conv. are expressed. This may be due to the weaker interaction between F$^-$ and TMA$^+$. Furthermore, changing KOH to NaOH also degrades the activities of water reduction and hydrogenation of **1a**. This is due to the larger $n$ and radii of the hydrate Na$^+$ (Na$^+$(H$_2$O)$_{13}$) than those of K$^+$ (K$^+$(H$_2$O)$_7$), which weakens the interaction between the halide anion and hydrated cation, thus resulting in a weaker ability to activate H$_2$O[41]. However, when using Co foil without a surface F$^-$ modifier as the cathode, no significant differences in HER and **1a** Conv. are observed after replacing KOH with TMAH or NaOH (Supplementary Fig. 16). These results may illustrate that the key role of surface F$^-$ is to promote H$_2$O activation via the interactions between F$^-$ and hydrated K$^+$.

The pH value also influences the activity and product distributions in electrochemical transformations. Therefore, we investigated the effect of pH on the electrocatalytic hydrogenation of **1a** using a mixed solution of dioxane with 1.0 M KOH, 0.5 M K$_2$CO$_3$, and 0.5 M K$_2$HPO$_4$, respectively, at −0.2 V vs. reversible hydrogen electrode (RHE) (Supplementary Fig. 17 and Table 2). Supplementary Fig. 17b shows that the Conv. of **1a** is positively related to the pH value, and 1.0 M KOH gives the best result. We rationalize that the better performance for KOH

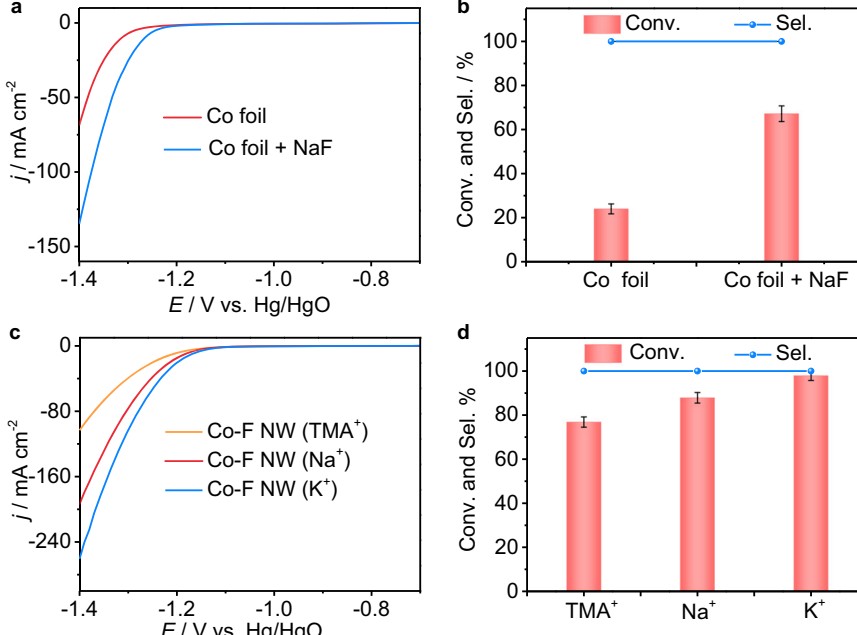

**Fig. 3 | The effects of fluorine and cations in the electrolyte on the electrocatalytic hydrogenation of 1a. a** LSV curves of Co foil recorded in 1.0 M KOH without **1a** at a scan rate of 10 mV s$^{-1}$, and **b 1a** Conv. and **2a** Sel. over Co foil with and without NaF. Reaction conditions: **1a** (0.1 mmol, 14.28 mmol L$^{-1}$), a mixed solution of 1.0 M KOH/dioxane (6:1 v/v, 7 mL), −1.2 V, 6 h, RT. **c** LSV curves of Co-F recorded in 1.0 M MOH (M = TMA$^+$, Na$^+$, and K$^+$) electrolyte without **1a** at a scan rate of 10 mV s$^{-1}$, and **d 1a** Conv. and **2a** Sel. over Co-F. Reaction conditions: **1a** (0.1 mmol, 14.28 mmol L$^{-1}$), a mixed solution of 1.0 M MOH (M = TMA$^+$, Na$^+$, and K$^+$)/dioxane (6:1 v/v, 7 mL), −1.1 V, 6 h, RT. Error bars correspond to the standard deviation of three independent measurements.

may be attributed to the production of more H* by accelerating the activation of H$_2$O (Supplementary Fig. 17a). These results may demonstrate that the electrocatalytic hydrogenation of **1a** easily proceeds at a higher pH value.

Next, some experiments were further performed to investigate the mechanism. Adsorption is usually an essential step in electrochemical transformations. First, to determine whether the electrocatalytic hydrogenation of **1a** occured in the bulk solution or on the surface of the Co-F electrode, 1-dodecanethiol was employed to modify the Co-F cathode. The conversion of **1a** is obviously decreased under the standard reaction conditions after introducing 1-dodecanethiol to the reaction system (Supplementary Fig. 18). This may reveal that the electrocatalytic hydrogenation of **1a** occurs mainly on the Co-F surface. We then investigated the adsorption modes of substrates on the Co-F surface (Fig. 4a). Approximately 34.5% and 32.8% conversions are achieved when using 2,8-dimethylquinoline **1r** and benzo[*h*]quinoline **1 s** as the substrates, respectively. Quinoline *N*-oxide **1t** still works well under standard conditions. These results may provide indirect support for the flat adsorption of quinoline in our reaction, agreeing with our DFT results (Fig. 1a). We have also investigated the interaction strengths of **1a** with Co and Co-F using the temperature-programmed desorption (TPD) technique as reported in the literature[36]. However, there is no difference in the desorption temperature of **1a** on either sample. We speculate that a thin layer of cobalt oxide may form on the Co-F surface during the transfer of the sample from the electrochemical cell for the TPD test. Additionally, electron-deficient *N*-heteroarenes are prone to accept an electron to generate the corresponding radical anion under cathodic reduction conditions[32]. To investigate the reduction behavior of **1a**, LSV curves were recorded in anhydrous acetonitrile (AN) containing tetrabutylammonium tetrafluoroborate (TBABF$_4$) as the electrolyte over an inert glass carbon cathode. Figure 4b demonstrates a distinct peak (red line) appearing at ~−2.0 V when **1a** was added to the system, revealing its reduction by accepting the electron. A positive shift of the

reduction peak (blue line) after further addition of H$_2$O confirms the promotion role of H$_2$O in **1a** reduction. Therefore, we deduce that the electrocatalytic hydrogenation of **1a** begins with **1a** gaining an electron, which is similar to Lei's work[32]. Furthermore, in situ-formed H* via H$_2$O dissociation is proven to be the key species in electrocatalytic transfer hydrogenation reactions[23–25,29]. Electron paramagnetic resonance (EPR) measurements with 5,5-dimethyl-1-pyrroline-*N*-oxide (DMPO) as the radical spin-trapping reagent confirm the generation of hydrogen radicals (Fig. 4c, marked by #, and Supplementary Note 5) during the reaction. After adding tertiary butanol (*t*-BuOH), which can scavenge hydrogen atoms to the system, the conversion of **1a** is significantly impeded (Fig. 4d and Supplementary Note 6), suggesting the necessity of hydrogen radicals for the hydrogenation of **1a**. Finally, no **2a** is detected when the electrocatalytic hydrogenation of **1a** is conducted in an anhydrous acetonitrile solution with electricity in the absence or presence of 1.0 atm H$_2$ (Supplementary Fig. 19). Consequently, the hydrogenation of **1a** should be solely due to generated H* from water dissociation.

A possible reaction mechanism is proposed (Fig. 4e and Supplementary Fig. 20). First, **1a** adsorbs on the Co-F surface through flat adsorption and accepts an electron to generate the radical anion. Then, it abstracts a proton from water to produce radical intermediate **I**, which quickly isomerizes to a more stable resonance structure **II** owing to the coexistence of benzyl and allyl radicals. Next, the C4 position of intermediate **II** couples hydrogen radicals, giving rise to partially hydrogenated product **III**. The Gibbs free energy of **III** is more negative than that of the other two isomers, which may suggest its easier formation (Supplementary Fig. 20). **III** experiences either a H radical addition pathway or a proton-coupled electron transfer process to deliver intermediate **IV**, which combines with another H radical to obtain product **2a**. Finally, **2a** desorbs from the Co-F surface to regenerate the catalytic sites for the next reaction cycle. The EPR results also reveal the formation of carbon radicals **I**, **II**, or **IV** (Fig. 4c, marked by *). Therefore, this electrocatalytic hydrogenation of **1a** may

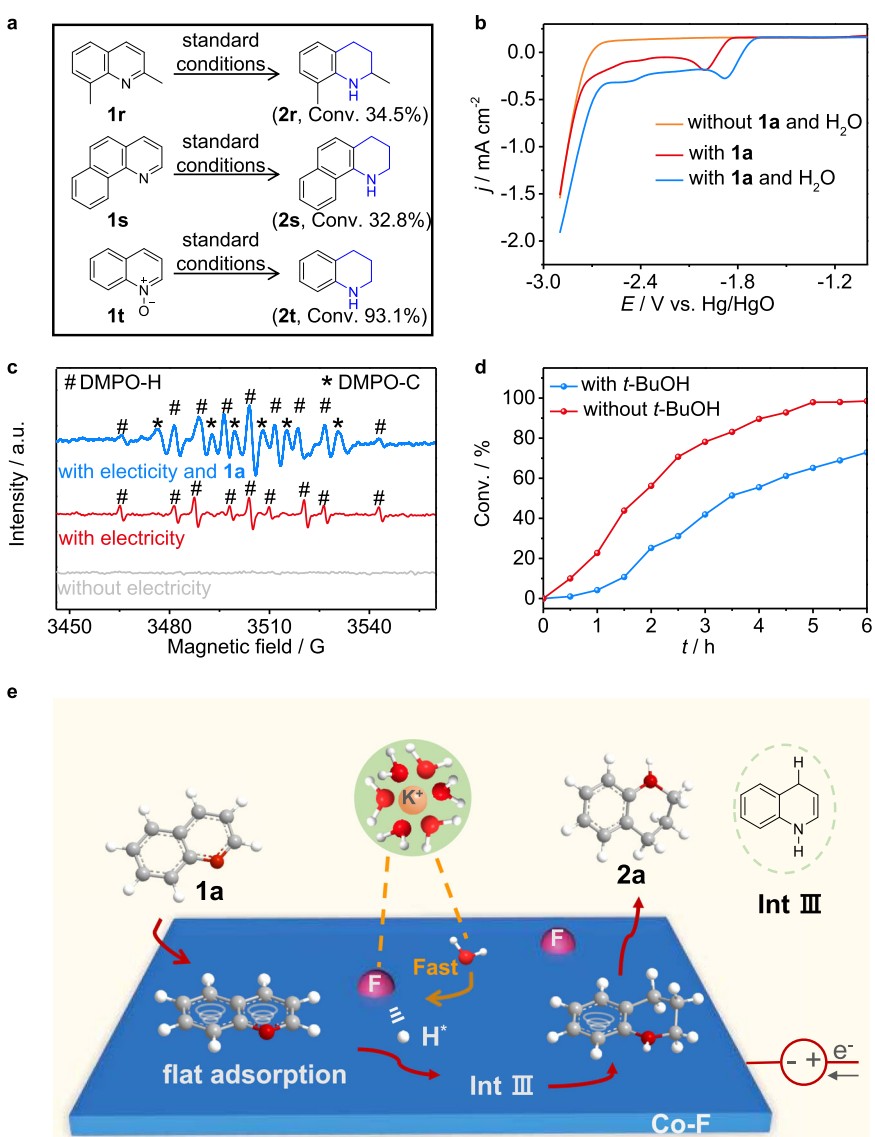

**Fig. 4 | Mechanistic studies of electrocatalytic hydrogenation of quinolines over a Co-F cathode. a** Control experiments: substrates (0.2 mmol, 28.57 mmol L$^{-1}$), Co-F electrode (working area: 1.0 cm$^2$), a mixed solution of 1.0 M KOH/dioxane (6:1 v/v, 7 mL), −1.1 V, RT, 12 h. **b** LSV curves in anhydrous acetonitrile at a scan rate of 10 mV s$^{-1}$ with and without 0.2 mmol **1a**. **c** Electron paramagnetic resonance trapping for hydrogen (#) and carbon (*) radicals over the Co-F catalyst. **d** Comparisons of **1a** conversion with and without *t*-BuOH. **e** Proposed reaction mechanism.

proceed through a 1,4/2,3-addition pathway involving H radicals, remarkably distinct from the often-mentioned 1,4/1,2-addition in thermal catalysis[5,6,9].

## Methodology universality and utility

The generality of this electrocatalytic hydrogenation of quinolines over a Co-F cathode is evaluated (Table 1 and Supplementary Note 7). Because the reduction potential and solubility of each substrate are different, a mixed solution of 1.0 M KOH/dioxane and a longer reaction time is used for some substrates. Quinoline and a variety of functionalized quinolines bearing electron-donating and electron-withdrawing groups on the benzene rings or pyridine rings can be transformed to the corresponding hydrogenated products with good to excellent conversions and moderate to high isolated yields (**2a-j**). To be specific, quinolines featuring methyl, methoxy, and amide groups in the 6-position all work well to deliver the corresponding THQ products (**2b-d**) in 85–90% isolated yields, and 6-fluoro-1,2,3,4-tetrahydroquinoline (**2 f**) is also successfully prepared with 80.5% conversion. The commonly challenging and versatile −Cl and −COOH

functionalities are well retained in the products (**2e** and **2 g**), providing rich opportunities for fabricating important complexes. Additionally, installing a −CH$_3$ substituent in either the 8- or 2-position of quinoline (**2 h** and **2i**) does not exert an influence on the reaction efficiency. Hydrogenation of the pyridine ring usually becomes more difficult toward substitution in the 4-position of the quinoline by using previous methods[12]. In this work, the hydrogenation of 4-methylquinoline can efficiently afford **2j** in a high yield. For the case of 1,5-naphthyridine, partially reduced **2k** as the sole product is still obtained. These satisfactory results reveal that neither the electronic nor the steric hindrance effects have noticeable influences on the electrocatalytic hydrogenation of quinolines over Co-F, demonstrating their general applicability. Furthermore, the fluorine effect on the electrocatalytic hydrogenation of other *N*-heteroarenes is also significant. For example, the conversions of quinoxaline and 6,7-dimethoxy-1-methyl-3,4-dihydroisoquinoline over the Co foil cathode are obviously enhanced after adding a small amount of NaF into the cathodic cell (Supplementary Fig. 21a, b). Thus, Co-F is used for the electrocatalytic hydrogenation of other *N*-heteroarenes, such as quinoxalines,

**Table 1 | Substrate scope of electrocatalytic hydrogenation of quinolines and other *N*-heterocycles with $H_2O$ over the Co-F cathode**

quinoline substrates[a]

**2a**, 98.2% (94%)   **2b**, 96.5% (85%)   **2c**, 97.3% (88%)   **2d**, 97.6% (90%)

**2e**, R=Cl, 93.2% (72%)[b]
**2f**, R=F, 80.5% (68%)[c]   **2g**, 89.6% (88%)[c]   **2h**, 90.6% (75%)[b,c]   **2i**, 97.3% (82%)[c]

**2j**, 96.8% (89%)   **2k**, 98.1% (89%)   **2l**, 95.8% (88%)   **2m**, 92.6% (78%)

other *N*-heterocycle substrates

**2n**, 87.5% (85)[c]   **2o**, 89.3% (87)[c]   **2p**, 96.5% (45)[d]   **2q**, 97.2% (87)[d]

[a]Reaction conditions: substrates (0.2 mmol, 28.57 mmol L[−1]), Co-F electrode (working area: 1.0 cm[2]), a mixed solution of 1.0 M KOH/dioxane (6:1 v/v, 7 mL), −1.1 V, RT, 12 h.
[b]A mixed solution of 1.0 M KOH/dioxane (5:2 v/v, 7 mL).
[c]−1.2 V, 18 h.
[d]3,4-Dihydroisoquinolines were used as substrates. Conv. of substrates and isolated yields of products are reported. Conv. was calculated by gas chromatography (GC) using dodecane as an internal standard. Isolated yields of products are given in round brackets.

isoquinolines, and 3,4-dihydroisoquinolines. As expected, they are all amenable to our strategy, producing hydrogenated products with high conversion and selectivity (**2l**–**2q**). Moreover, to further investigate the chemoselectivity of this electrocatalytic hydrogenation reaction, we also examined quinoline substrates containing readily reducible functional groups, such as −C≡CH, −CN, and −CHO, under our reaction conditions. Unfortunately, those fragile groups have difficulty surviving, and a mixture of hydrogenated products is obtained (Supplementary Figs. 22–24). Therefore, precise screening of reaction conditions or further modification of the electrocatalyst to improve the compatibility of such more readily reducible functional groups will be highly needed in future work. Interestingly, when the Co foil cathode is decorated by additional F[−], the conversion of the electrocatalytic hydrogenation of 4-ethynylaniline is markedly boosted (Supplementary Fig. 21c), further demonstrating the generality of the fluorine effect in improving the electrochemical performance of other organic reactions.

The utility of our method is further demonstrated. Deuterium-labeling offers an important tool for drug development, the mechanistic study of organic reactions, etc[49–51]. Therefore, the first application involves the synthesis of deuterated analogs of THQs. By employing safe and inexpensive $D_2O$ as the deuterated source, THQs containing different amounts of D atoms are facilely prepared with high yields and deuterated ratios (Fig. 5a), avoiding the use of other expensive and hard-to-obtain deuterated reagents. Second, our reaction can be easily scaled up. A total of 1.2 g of **1a** (96.2% Conv.) and 1.2 g of **1b** (92.5% Conv.), which are important building blocks for fabricating bioactive compounds, can be expediently synthesized over an enlarged Co-F cathode at −1.2 V (Fig. 5b and Supplementary Fig. 7b and Note 8). Impressively, hydrogenation of **1a** can also be implemented under galvanostatic conditions. No decreases in the conversion and selectivity are observed even at a current of 100 mA cm[−2] (Supplementary Fig. 25), demonstrating the good flexibility of our method. Third, developing a thermodynamically more favorable organic oxidation reaction to replace the low-value and kinetic-sluggish oxygen

evolution reaction (OER) is of great significance to enhance the performance of cathodic reactions in an aqueous solution[52–56]. Electrocatalytic hydrogenation of **1a** can be accomplished by adopting a divided Co-F||NiSe two-electrode electrolyzer by using the oxidation of 1,6-hexanediamine (**4a**) to replace OER. Adiponitrile (**5a**), an important industrial feedstock for nylon production, and **2a** with high yields are simultaneously synthesized. To achieve a benchmark current density of 30 mA cm[−2], nearly 320 mV voltage is saved (Fig. 5c and Supplementary Note 9), showing good promise. The highly selective synthesis of tetrahydroquinoline paired with the preparation of adiponitrile at a lower energy input illustrates the advantage of our method, which is otherwise difficult to access by traditional hydrogenation methods of quinoline.

## Discussion

In summary, we report the efficient electrocatalytic hydrogenation of quinolines to THQs by using clean and abundant water as the hydrogen source over a fluorine-modified cobalt NW cathode under ambient conditions, forming a sharp contrast to currently reported methods (Supplementary Table 3). This strategy shows high efficiency and excellent regioselectivity toward a wide range of functionalized quinolines and other *N*-heteroarenes. Theoretical results reveal that surface F[−] enhances the adsorption of substrates and boosts the activation of $H_2O$ to H*, contributing to the superior performance of Co-F. A 1,4/2,3-addition pathway involving hydrogen radicals is proposed for this transformation. Additionally, the potential utilities are impressively demonstrated by the easy and effective preparation of deuterated THQs analogs and the gram-scale synthesis of bioactive precursors. Furthermore, hydrogenation of quinoline can be well achieved in a two-electrode electrolyzer to generate value-added adiponitrile at the anode with lower energy input, further confirming the significance of this electrochemical method. Moreover, the fluorine modification strategy can be extended to other organic reactions to improve their electrochemical performances, showing good generality. Our work not only provides an efficient and green approach for the selective

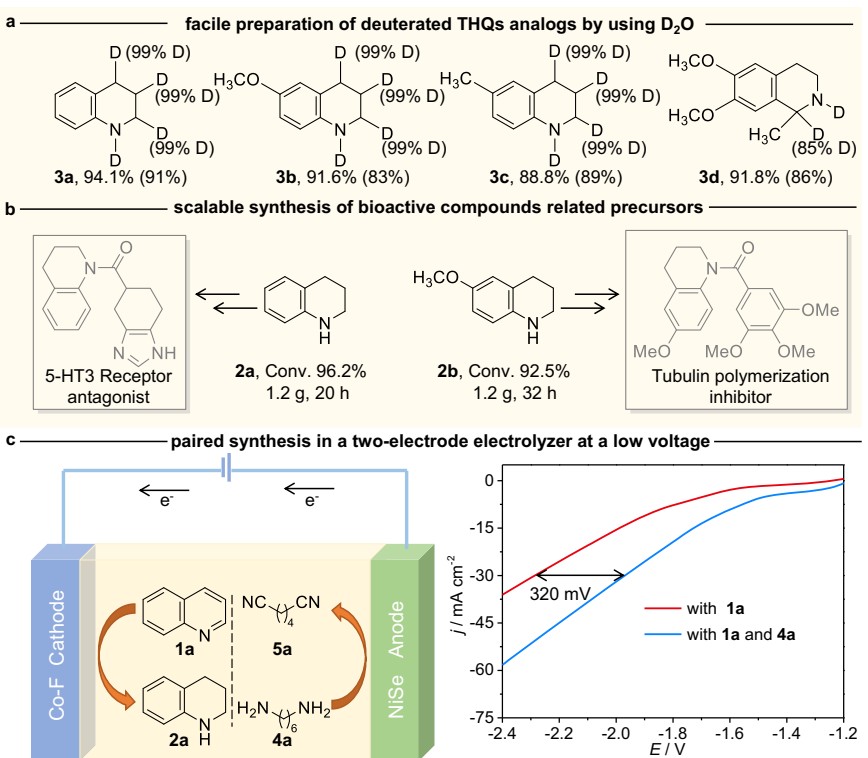

**Fig. 5 | The applications of the methodology. a** Facile preparation of deuterated analogs of THQs by using D$_2$O over the Co-F cathode. Reaction conditions: substrates (0.2 mmol, 28.57 mmol L$^{-1}$), Co-F electrode (working area: 1.0 cm$^2$), a mixed solution of 0.5 M K$_2$CO$_3$ (D$_2$O)/dioxane (6:1 v/v, 7 mL), −1.1 V, RT, 16 h. **b** Gram-scale synthesis of bioactive precursors over an enlarged Co-F (3.0 × 3.0 cm$^2$) cathode. **c** Paired synthesis of THQ and adiponitrile in a Co-F||NiSe two-electrode electrolyzer.

hydrogenation of quinolines but also offers a paradigm for designing highly active metal electrocatalysts with surface adsorbates via electrochemical treatment for other electrochemical transformations (e.g., hydrogenation of nitrate to ammonia, oxygen reduction to hydrogen peroxide, and electrooxidation of halogen anions to hypohalite).

# Methods

## Synthesis of Co(OH)F Nanowire (NW) on nickel foam (NF) support

The Co(OH)F/NF was synthesized through a simple hydrothermal process according to a modified reported method[57]. First, Co(NO$_3$)$_2$·6H$_2$O (0.582 g), CO(NH$_2$)$_2$ (0.60 g) and NH$_4$F (0.296 g) were added to 35 mL deionized water (DIW). After ultrasonication for 20 min, the homogeneous solution was transferred into a 50 mL Teflon-lined stainless autoclave, and freshly treated NF (4.0 × 3.0 cm$^2$) was immersed into the solution. The autoclave was sealed and heated at 120 °C for 10 h in an oven. After the autoclave was cooled to room temperature naturally, the Co(OH)F/NF was removed and washed with DIW and ethanol. The final product was dried in a 60 °C vacuum oven overnight for use.

## Synthesis of Co-F NW via electroreduction of Co(OH)F NW precursor

The Ivium-n-Stat electrochemical workstation (Ivium Technologies B.V.) is used for the electroreduction of Co(OH)F NW to Co-F NW. Co(OH)F NW with a working area of 1.0 cm$^2$ was used as the working electrode. A Hg/HgO (1.0 M KOH) and a carbon rod were used as the reference electrode and the counter electrode, respectively. Electrochemical activation was carried out in a typical three-electrode system with 1.0 M KOH as the electrolyte at −1.2 V vs. Hg/HgO until the *i-t* curves become stable.

## Characterizations

Scanning electron microscopy (SEM) images were taken with a FEI Apreo S LoVac scanning electron microscope. Transmission electron microscopy (TEM) images were obtained with a FEI Tecnai G2 F20 microscope. The X-ray diffraction (XRD) patterns of the products were recorded with a Rigaku Smartlab 9KW diffraction system using a Cu K$\alpha$ source ($\lambda$ = 0.15406 nm). X-ray photoelectron spectroscopy (XPS) measurements were performed on a Thermo Fisher Scientific ESCALAB-250Xi spectrometer. All the peaks were calibrated with the C 1$s$ spectrum at a binding energy of 284.8 eV. The in situ Raman spectra were recorded on a Renishaw inVia reflex Raman microscope under an excitation of 532 nm laser light with a power of 20 mW. X-ray absorption spectroscopy (XAS) measurements were undertaken at the 1W1B beamline of the Beijing Synchrotron Radiation Facility (BSRF). The XAS spectra were analyzed with the ATHENA software package. To exclude the interference of the NF substrate, Co(OH)F powders were stripped off from the NF substrate and loaded on CP for in situ Raman and in situ XAS tests. Nuclear magnetic resonance (NMR) spectroscopy was recorded on a JEOL JNM-ECZ400S/L1 instrument at 400 MHz ($^1$H NMR), 101 MHz ($^{13}$C NMR), and 376 MHz ($^{19}$F NMR). Chemical shifts were reported in parts per million (ppm) downfield from internal tetramethylsilane. Multiplicity was indicated as follows: s (singlet), d (doublet), t (triplet), q (quartet), m (multiplet), dd (doublet of doublet), br (broad), and dt (doublet of triplet). Coupling constants were reported in hertz (Hz). Qualitative analysis of samples was performed using gas chromatography-mass spectrometry (GC-MS, Agilent 8860 A GC-5977 MS) equipped with a HP-5MS column (30 m × 0.25 mm) and electron ionization. Quantitative analysis of samples was conducted by gas chromatography (GC, Agilent 7890 A) equipped with a HP-5 column (30 m × 0.25 mm), thermal conductivity (TCD), and a flame ionization detector (FID). The injection temperature was set at 300 °C. Nitrogen was used as the carrier gas at 1.5 mL min$^{-1}$. High-resolution

mass spectrometry (HR-MS) was performed on a Thermo Scientific Q Exactive. Electron paramagnetic resonance (EPR) spectra were recorded with a Bruker EMX plus-6/1 spectrometer. All reported data are averages of experiments performed at least three times.

## General procedures for electrochemical measurements

Electrochemical measurements were carried out in a divided three-electrode electrochemical cell. Co-F with an exposed surface area of $1.0\ cm^2$ and a mass loading of ~$5.0 \pm 0.1\ mg\ cm^{-2}$ was used as the working electrode, and a carbon rod and Hg/HgO were used as the counter electrode and reference electrode, respectively. First, after Co-F was formed in situ, 0.1 mmol of quinolines dissolved in 1.0 mL dioxane was rapidly added to the cathode containing 6.0 mL 1.0 M KOH (pH = 13.6)[58] as the electrolyte and stirred to form a homogeneous solution. Then, chronoamperometry was carried out at a given constant potential of −1.1 V vs. Hg/HgO and stirred until the substrates were depleted. After the reactions were finished, dodecane was added to the reaction system as an internal standard. Then, the solution was extracted with ethyl acetate (EA). The EA phases were dried over $Na_2SO_4$, and the solvent was removed under reduced pressure. The residuals were separated either by flash column chromatography or using a thin-layer chromatography (TLC) plate to give the isolated yields or were analyzed by gas chromatography (GC) to provide the GC yields. The GC yields were determined by using dodecane as an internal standard. Additionally, the procedures of the deuterium-labeling experiments were similar to the electrochemical hydrogenation of quinolines, except that $H_2O$ and 1.0 M KOH were changed to $D_2O$ and 0.5 M $K_2CO_3$, respectively. Furthermore, the electrochemical hydrogenation of quinoline **1a** over other electrode materials was similar to that of the Co-F cathode. All the potentials in this work were referred to Hg/HgO without *iR* correction unless otherwise stated. All experiments were carried out at room temperature (25±0.5 °C).

## Product quantifications (1a selected as the example)

The conversion (Conv., %), selectivity (Sel., %), isolated yield (Yield, %), and deuterated ratio (%) were calculated using Eqs. (1) to (4):

$$\text{Conv. (\%)} : = \frac{n(\text{the comsumed quinoline substrate})}{n(\text{the initial quinoline substrate})} \times 100\% \quad (1)$$

$$\text{Sel. (\%)} : = \frac{n(\text{the obtained THQ product})}{n(\text{the consumed quinoline substrate})} \times 100\% \quad (2)$$

$$\text{Yield (\%)} : = \frac{n(\text{the obtained THQ product})}{n(\text{the added quinoline to the reactor})} \times 100\% \quad (3)$$

$$\text{Deuteration ratio (\%)} = \left(1 - \frac{\text{the residual integral}}{\text{the number of labeling sites}}\right) \times 100\% \quad (4)$$

## Electrochemical in situ Raman spectroscopy measurements

In situ Raman spectra were recorded on a Renishaw inVia reflex Raman microscope under excitation with a 532 nm laser under controlled potentials by an electrochemical workstation. The electrolytic cell was made up of Teflon with a thin round quartz glass plate as a cover to protect the objective. The working electrode was inserted through the wall of the cell to keep the plane of the working electrode perpendicular to the incident laser. Pt wire as the counter electrode was rolled to a circle around the cell. The Hg/HgO electrode with an inner reference electrolyte of 1.0 M KOH was used as the reference electrode. The spectrum was recorded every 30 min at −1.2 V vs. Hg/HgO with 1.0 M KOH as the electrolyte.

## Electrochemical in situ XAS measurements

The in situ electrochemical XAS at the Co K-edge was carried out at the 1W1B beamline of the BSRF. The electrolytic cell was made in-house with a Teflon containing 1.0 M KOH electrolyte, in which a graphite rod and Hg/HgO electrode were used as the counter electrode and reference electrode, respectively. The in situ XAS measurement was performed in fluorescence mode at a constant of −1.2 V vs. Hg/HgO to maintain the reduction situation. The photon energy was calibrated with the first inflection point of the Co K-edge in Co metal foil. The as-obtained XAS spectra were analyzed with the ATHENA software package.

## Computational details

Here, all calculations are performed using the Vienna ab initio simulation package (VASP) based on density functional theory (DFT)[59]. Projector augmented wave (PAW) pseudopotentials are employed[60]. The exchange-correlation contributions to the total energy are estimated by the generalized gradient approximation (GGA) with the Perdew−Burke−Ernzerhof (PBE) form[61]. The Hubbard U approach (DFT+U) is adopted to better describe the on-site Coulomb correlation of the localized 3d electrons for Co with $U - J = 3.42\ eV$[62,63]. An empirical dispersion-corrected DFT method (DFT-D$_3$) is carried out to reasonably describe the weak long-distance van der Waals effects[64]. The solvent effect ($G_{solv}$) is considered for thermodynamic corrections by using VASPsol[65]. The kinetic energy cutoff for plane-wave expansion is set to 500 eV. An energy convergence threshold of $10^{-4}$ eV is set in the self-consistent field iteration. The geometry optimization within the conjugate gradient method is performed with forces on each atom <0.05 eV Å$^{-1}$. A $p$ (6 × 6) slab model with three atomic layers is adopted to simulate the Co (111) surface. A vacuum layer of 15 Å is inserted along the $c$ direction to eliminate the periodic image interactions. The F-doped Co surface is modeled with one F atom being adsorbed on the pure Co surface. The bottom atomic layer is fixed, while other layers and the adsorbates are fully relaxed during structural optimizations. The Brillouin zone is sampled by a $k$-point mesh of 4 × 4 × 1. The reaction free energy change can be obtained with the following equation:

$$\Delta G = \Delta E + \Delta E_{ZPE} - T\Delta S$$

where $\Delta E$ is the total energy difference between the products and the reactants of each reaction step, and $\Delta E_{ZPE}$ and $\Delta S$ are the differences in zero-point energy and entropy, respectively. The zero-point energies of free molecules and adsorbates are obtained from vibrational frequency calculations. The free energy change of each step that involves electrochemical proton-electron transfer is described by the computational hydrogen electrode (CHE) model proposed by Nørskov et al.[66]. In this technique, zero voltage is defined based on the reversible hydrogen electrode, in which the reaction is defined to be in equilibrium at zero voltage, at all pH values, at all temperatures, and with $H_2$ at 101,325 Pa pressure. Therefore, in the CHE model, the free energy of a proton-electron pair is equal to half of the free energy of gaseous hydrogen at a potential of 0 V.

## Data availability

The data that support the plots within this paper are available from the corresponding author upon reasonable request. Source data are provided with this paper.

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

## Acknowledgements

We acknowledge the National Natural Science Foundation of China (Nos. 21871206 and 22001192) for financial support. We appreciate Dr. Shibo Xi at the National University of Singapore for the XAS discussion. We are also grateful for the kind support of Dr. Lirong Zheng at the 1W1B beamline of the Beijing Synchrotron Radiation Facility (BSRF).

## Author contributions

C.L. and B.Z. conceived the idea and directed the research. S.G., C.L., and B.Z. designed the experiments. S.G. and Y.W. performed materials synthesis and electrochemical experiments. S.G., Y.W., and C.L. ana-lyzed the NMR data. C.W. contributed to the density functional theory calculations. M.L. and Y.G. assisted with material characterization and product purification. C.L. wrote the paper. B.Z. revised the paper with comments from all authors.

## Competing interests

The authors declare no competing interests.
