## [Peer Review File · Nature Communications]

Title: Electrocatalytic hydrogenation of quinoline with water over a fluorine-modified cobalt catalystREVIEWER COMMENTS

Reviewer #1 (Remarks to the Author):

This manuscript related the chemical transformations that is an example of paired electrosynthesis. So, there is a need of adding discussion in the manuscript about paired electrolysis and its advantages of energy efficiency (ACS Sustainable Chem. Eng. 2021, 9, 18, 6148–6169, <https://doi.org/10.1021/acssuschemeng.1c00665>; Chem. Rec.. <https://doi.org/10.1002/tcr.202100047>). Although this manuscript have value based on higher selectivity of chemical's transformation, but is more suitable for the specialized Journal related process chemistry and engineering where synthesis of organic compounds is possible under environmentally benign conditions, instead Nature Communication's audience.

Reviewer #2 (Remarks to the Author):

Combing electrocatalytic HER with hydrogenation reaction to perform organic synthesis is significant and of apparent interest. The work by Zhang and Liu reports electrocatalytic transfer hydrogenation of quinolines with H₂O using fluorine-modified Co-F nanowires as the cathode. This system exhibits high reaction selectivity in the catalytic reduction of a wide range of quinoline derivatives to THQs. To confirm that water serves as the dihydrogen source, D₂O was used for the electrochemical reduction of 2a-2d, and the corresponding deuterated THQ isotopomers were obtained in excellent yields. DFT studies and electrochemical analysis suggest that coating the Co catalyst with F- enhances the absorption of the substrate at the electrode surface for the hydrogenation, which is in a 1,4/2,3-reducing sequence via H atom transfer. The scalable electrochemical synthesis of 2a and 2b by the present method is impressive and represents a promising application. Given the importance and novelty, I recommend this work to publish in Nature Communications once the following issues are adequately addressed.

To detail my recommendation

- 1) They need to rewrite the introduction and incorporate the first paragraph of "Consideration Factors" (page 4, line 67 to line 77) into the introduction part.
- 2) In the abstract, "using a safe and clean hydrogen donor over a cost-effective catalyst" is a confusing expression. Does it mean "hydrogen donor" vs. "cost-effective catalyst"? The whole sentence should be reorganized.
- 3) Does "active atomic hydrogen (H*)" mean "H atom" or "H radical"? It should be appropriately defined.
- 4) Page 2, change "promising potential" to "promising application".

- 5) Page 3, correct “hydrogenating method” to “hydrogenation method”.
- 6) “Furthermore, phenyl ring is more prone to lose rather than to obtain an electron” can be simplified to “phenyl ring is prone to reduction before...”
- 7) page 5, “Our data are correctly simulated using Co-Co and Co-O two paths”, how did they know their simulation is correct?
- 8) Descriptions for some experimental details are necessary. I suggest they add the details about how the yields and selectivity were determined in the Supporting Information.
- 9) ²H NMR spectra of 3a, 3b, 3c must be provided to compare with the corresponding ¹H NMR spectra.
- 10) They claimed that “the Conv. of 1a is positively related to the pH value, and 1.0 M KOH gives the best result”. However, the pH values for 1.0 M KOH, 0.5 M K₂CO₃, and 0.5 M Na₂SO₄ were not given.
- 11) For “the paired synthesis of THQ and adiponitrile using a Co-F || NiSe two-electrode electrolyzer”, does 1,6-hexanediamine serve as the hydrogen source in the hydrogenation of 2a?

Reviewer #3 (Remarks to the Author):

This manuscript describes the preparation of a surface-fluorine-modified cobalt (Co-F) nanowire catalyst via an electroreduction process, and its application for the electrochemical transfer hydrogenation of quinolines to tetrahydroquinolines (THQ) by using H₂O as the hydrogen source. The catalyst displays good recyclability and its performance has been tested for the reduction of different N-heteroarenes, including the synthesis of deuterated analogues. Moreover, the synthetic value of this methodology has been further demonstrated by synthesizing bioactive precursors in a preparative scale, as well as through a paired synthesis of THQ and adiponitrile, this latter by electrochemical oxidation of 1,6-hexanediamine. With regard to the catalyst characterization, the wide range of techniques used, such as SEM, (HR)TEM, EDS, XRD, XPS, Raman XANES, and XFAFS, have revealed that the electroreduced catalyst Co-F consists on a nanowire composed by a non-reduced Co(OH)₂ core coated by fluorine-adsorbed amorphous metallic cobalt nanosheets. Particular emphasis has been placed on the elucidation of the role of the fluorine-modified surface on the electroreduction process. Based on the novelty of the synthetic strategy, the “a priori” impressive catalytic results, and the conceptual advance presented in this work, it might be suitable for Nature Communications. However, in my opinion, this version is far to be acceptable for such a world-class journal.

- 1) Nowadays, the hydrogenation of quinolines to THQ catalyzed by non-noble metal-based catalysts is well known. However, the sentence “Additionally, noble metal and high temperature are always

required to activate the hydrogen sources and/or quinoline substrates,^{5,9} which increase the possibility of undesirable hydrogenation of phenyl ring.¹²⁻¹³ seems to express the opposite. Representative works, such as the ones presented in Supplementary Table 2, and other important ones (DOI: 10.1039/C8SC02744G; DOI: 10.1021/acscatal.7b04260) should be mentioned in the introduction. In addition, reference 12 is not correctly used, since this does not reflect the over-hydrogenation of quinolines.

2) No electroreduction of quinolines is presented in reference 30. Therefore, the sentence “... However, conversion yields of quinolines are low.” needs to be rewritten.

3) Authors commented that carried out the optimization of different adsorption modes of quinoline on the catalyst surface, and the flat adsorption resulted to be the most favored. These comparative theoretical results should be included in the supporting information.

4) The theoretical results predict that the presence of fluorine atoms on the catalyst surface, significantly enhances the adsorption of quinoline, and boosts the electrochemical water reduction. Both should be corroborated by experimental results. However, whereas the latter prediction has been demonstrated by complementary catalytic studies with fluorine-free related materials, no experimental evidences on the different adsorption strength are reported.

5) To avoid readers' confusion, graphics a) and b) in Figure 1 should be represented in the same adsorption energy scale.

6) According to the authors' statement, “The exposing metallic Co works as the active center for the electrochemical transfer hydrogenation reaction of quinolines.” To confirm this hypothesis, the electrocatalytic reaction in the presence on the non-activated Co(OH)F should be tested. If this is the case, since similar cathodic potential is used (around -1.2 V) for both processes, i.e. catalyst activation and catalytic reaction, an induction period in the formation of THQ should be observed.

7) The term “conversion yield”, used along the manuscript, leads to confusion. Typically, “conversion” is referred to the substrate (i.e. quinoline), while the term “yield” is used for products.

8) There is no clear information on the quantification methods used to determine conversion and selectivity. From the equations shown in the methods section, it seems both have been calculated by GC. However, no internal standard were added to the reaction mixture. Actually, it is a highly unreliable method because side products are easily overlooked and underestimated. Moreover, comments on Figure 2b mention that selectivity (99%) is determined according to Supplementary Note 3, which refers to GC-MS measurements, being this a qualitative rather than a quantitative tool.

9) The F 1s XPS spectrum of the reused catalyst shown in Supplementary Fig. 8 does not reflect the loss of surface fluorine species. For that, the surface atomic ratio Co/F should be calculated and compared with that of the fresh catalyst (i.e. with that one just obtained after the electrocatalytic activation).

Could be possible to reactivate the reused catalyst after eight runs?

10) In the mechanism studies, authors state that "... we observe a lower onset potential and a significantly enhanced Conv. of 1a after adding NaF into the electrolyte over a Co foil cathode (Figs. 3a-b)." On the contrary, Fig. 3a shows the opposite. These results seem to conflict with each other, why?

11) Authors carried out a study depending on the pH by using different electrolytes, such as KOH, K₂CO₃, and Na₂SO₄. However, they previously remark the importance of the cation in the formation of a specific anion-hydrated cation pair, which seems to be crucial for accelerating H₂O activation. Therefore, to rule out the effect of the cation in the pH variation study, the same cation should be used for all experiments. That is, Na₂SO₄ needs to be substituted by K₂SO₄. On the other hand, pH values should be included in Supplementary Figure 11a.

12) Authors should explain in more detail why they think that the use of the quinoline derivatives 1q, 1r, and 1s are good candidates to discern between the different adsorption modes of quinolines.

13) In Table 1, isolated yields of the obtained products should be included. In addition, it would be interesting to further investigate the chemoselectivity of this electrochemical reduction process by testing N-heterocycles containing other reducible functional groups, such as double and triple C-C bonds, ketone, aldehyde, nitrile, ester, and amide groups.

14) In general, for the convenience of the readers, the reactions conditions used in each of the experiments shown in Figures should be indicated in the Figure caption.

Reviewer #4 (Remarks to the Author):

Summary: Fluorine-modified cobalt (Co-F) nanowires were synthesized and used for the electrochemical hydrogenation of quinolines. The cobalt with metallic character and fluorine present was found to both be key to the successful hydrogenation of the quinolines. In alkaline conditions of KOH, the reaction proceeded without limitations, able to achieve 100% conversions and 99-100% selectivities. No unsaturated ring hydrogenation was observed.

Comments:

- Electrochemical hydrogenation is advantageous for selective hydrogenation, particularly when ring saturation is undesired. The catalyst proposed in the work holds significant promise for its selectivity and activity.

The work could be enhanced through how it is presented and changing at what point experiments are compared to each other.

- While it is great to show that 100% conversions can be reached, illustrating that the reactions are not equilibrium limited, catalytic comparisons at or near 100% conversion have diminished value. Catalytic studies to allow for distinguishing of catalysts and conditions are better done far away from 100% conversion. This is because 100% conversion results do not distinguish how long it took to get to 100% conversion – has the system been sitting there for a few hours or just a few minutes? This is especially true in the catalyst recycle results. The catalyst is likely deactivating after each run, though only at run 7 and beyond is the deactivation significant enough to slow the reaction down to not be able to go to completion in the given time.
- In the discussion of pH, the actual solution pHs should be reported (for the whole solution, not just components of the solution). It is also not clear if the pH studies are actual anion effect studies. The anion can also have an impact on the electrochemical performance – perhaps the anion is replacing F- for example. This may be independent of the pH.
- pH studies should be reported vs RHE and studied at constant overpotential. As presented, the overpotentials are dramatically different from each other. Even when the same electrolyte and catalyst were used, the overpotentials can lead to different results (ex. Fig 2 a)
- The experiment with 1atm H₂ is a good start at showing that the electrochemical hydrogenation is not from H₂ produced via HER that is then further used for hydrogenation. The experiment is incomplete, however. This is because the cobalt can be partially oxidized in the KOH when at open circuit potential. When a reducing potential is applied, the cobalt is reduced. This means that the catalyst being studied at open circuit potential with H₂ gas is not the same catalyst and would not have the same activity as the one being studied in electrochemical hydrogenation. A good follow-on is to do the electrochemical hydrogenation with 1 atm H₂ being sparged through the solution. Compare this to the electrochemical hydrogenation experiment without H₂ to prove that H₂ is not the reactant.
- Nickel foam is used as the support for the catalyst. No mention of any role of nickel is mentioned in the study and if it has any catalytic role.
- It is not clear if the electrochemical system studied is homogeneous or biphasic. Mixing is said to lead to a homogeneous solution though an aqueous layer is also formed. This is confusing and should be clear up front.
- 1.0M KOH or other molarities are actually not used in the studies. Additional solvent and reactant is added to a solution already containing 1.0M KOH. This dilutes the base (or other electrolyte). The concentrations used in the electrochemistry, not in solutions prior to being combined should be used and reported.
- Concentrations, not moles should be reported for the quantities of reactants used. This allows the work to be much more transferrable.
- ‘electrochemical transfer hydrogenation’ is awkward and not customarily used. Rather, ‘electrochemical hydrogenation’ is usually the phrase used for the reactions being studied.
- The title is also awkward and lacks information. Cobalt is also key. Perhaps ‘Fluorine-modified cobalt catalysts for electrochemical hydrogenation of quinolines at room temperature’ or something like this would be better and more informative
- Justification of doing electrochemical hydrogenation because hydrogen is a safety hazard is difficult to make because the petrochemical industry safely uses high pressure hydrogen regularly. Rather, focusing on selectivity advantages and decarbonization strategies would be more convincing.

- Line 72-73. Stating that Co-based cathodes are widely used in hydrogen production is misleading. Commercial HER catalysts do not involve Co. Recommend instead of widely used stating widely studied.
- Figure 1 caption. Reword to emphasize the synthesis of Co-F NWs. It is easy to miss that the Co-F was electrochemically synthesized so the caption gets confusing. Also, by having the caption clearer, the message about how Co-F was synthesized becomes clearer. 'Synthesis of Co-F NWs by electroreduction of Co(OH)F NWs'
- Fig. 2 & 3 captions. Conditions need to be stated including reaction duration times and electrolytes.
- Make sure to define "NF" in text and experimental sections. While this stands for 'nickel foam' it could just as easily stand for 'nanofiber'
- Line 197. Provide a reference for the non-covalent Coulomb interaction.
- Line 202. Define what 'n' is
- Line 242. 'heavily dragged' is a strange phrase
- A table of contents for the supplemental information would be appreciated

Jan. 24, 2022

A point-by-point response to the reviewers' comments

To reviewer 1:

Reviewer letter: This manuscript related the chemical transformations that is an example of paired electrosynthesis. So, there is a need of adding discussion in the manuscript about paired electrolysis and its advantages of energy efficiency (*ACS Sustainable Chem. Eng.* 2021, 9, 18, 6148–6169, <https://doi.org/10.1021/acssuschemeng.1c00665>; *Chem. Rec.* <https://doi.org/10.1002/tcr.202100047>). Although this manuscript have value based on higher selectivity of chemical's transformation, but is more suitable for the specialized Journal related process chemistry and engineering where synthesis of organic compounds is possible under environmentally benign conditions, instead Nature Communication's audience.

Answer: We appreciate the reviewer for giving a positive comment on our work "*this manuscript have value based on higher selectivity of chemical's transformation*". However, for the reviewer's other comments, we need to explain them as follow.

First, we do not agree with the reviewer that this manuscript is an example of paired electrosynthesis.

The focus of our manuscript is to ***provide a mild and efficient approach for selective hydrogenation of N-heteroarenes under ambient conditions*** rather than reporting a paired electrosynthesis. Selective hydrogenation of quinolines has been a long-standing subject in synthesis chemistry, which provides a straightforward and convenient route to produce a variety of fine chemicals and drug-related skeletons. Despite impressive achievements, the dominant thermo-catalytic methods are mainly relying on using of high-pressure H₂ (from a few to several tens of atmospheres) and high temperature (usually ≥ 100 °C) to obtain satisfactory reaction efficiency. These usually not only need special apparatus with cautious operations, but also bring about severe concerns on the cost, safety risks, as well as product selectivity and functional group compatibility, heavily impeding the practical applications. Therefore, ***developing a mild and efficient approach method for hydrogenation of quinolines, especially using a cheap and easy-to-handle hydrogen donor catalyzed by a cost-effective catalyst, is highly significant.***

Electrochemical hydrogenation by using water as the hydrogen source is becoming a promising approach in the synthetic field. Current studies are mainly focused on the hydrogenation of NO₃⁻, and other easily reducible organic substrates (e.g., nitros, aldehydes, ketones, alkynes, and halides). However, electrochemical hydrogenation of quinolines with high resonance stability has been rarely touched and

still faces a technological challenge.

In our work, a cobalt cathode with surface fluorine (Co-F) is synthesized via electroreduction of Co(OH)F nanowire precursors, which shows high activity for electrochemical hydrogenation of quinolines to selectively synthesize tetrahydroquinolines (THQs) by using water as the hydrogen source at room temperature. The significances of our manuscript are listed below:

(1) An efficient electrochemical hydrogenation of quinolines with water over a surface F adsorbed Co cathode (Co-F) at RT.

- ✓ A RT electrochemical hydrogenation of quinolines with water is achieved with up to **99% conversion of quinolines and 99% selectivity of THQs** over Co-F.
- ✓ Hydrogenation reactions **only occur at N-heterocycle moieties**, revealing **excellent regioselectivity**.
- ✓ Co-F cathode can **be reused for seven runs** without obvious decreases in the conversion yield and selectivity, showing good stability.

(2) Unveiling high-performance origin, and proposing a distinct 1,4/2,3-addition pathway involving hydrogen radicals.

- ✓ Theoretical calculations reveal **surface F enhances the adsorption of quinolines and promotes H₂O activation** to form active hydrogen (H^{*}), conducting to the hydrogenation of quinolines.
- ✓ **F⁻-K⁺(H₂O)₇ networks may be formed** to facilitate the production of H^{*} via H₂O reduction, and K⁺ and a high pH value are also very significant to obtain a good performance.
- ✓ **Flat adsorption** of quinoline is proved, and **a 1,4/2,3-addition pathway involving hydrogen radicals** is proposed, distinct from the oft-mentioned 1,4/1,2-addition in thermal catalysis.

(3) Outstanding methodology universality and promising utility.

- ✓ **20 examples** of quinolines and other N-heteroarenes are hydrogenated.
- ✓ **Expedient deuterium introduction** by using D₂O shows a great promise for deuterated synthesis, and **gram-scale fabrication** of bioactive precursors demonstrates the practical utility.
- ✓ **Paired synthesis** of tetrahydroquinoline and adiponitrile in a divided two-electrode electrolyzer at a lower voltage makes our strategy stand out from the current approaches.

Thus, our work ***not only provides a mild and efficient approach for selective hydrogenation of N-heteroarenes at ambient conditions, but also offers a paradigm for designing highly active metal electrocatalysts with surface adsorbates via electrochemical treatment synthesis for other electrochemical transformations*** (e.g., hydrogenation nitrate to ammonia, oxygen reduction to hydrogen peroxide, and electrooxidation of halogen anion to hypochlorite) to produce industrially important feedstock.

Furthermore, we respect the reviewer's suggestion to add discussions about paired electrolysis and cite the suggested references in the reference section (Refs. 52 & 53) in the revised manuscript. The

descriptions can also be found as follow: “*Thirdly, developing a thermodynamically more favorable organic oxidation reaction to replace the low-value and kinetic-sluggish oxygen evolution reaction (OER) is of great significance to enhance the performance of cathodic reactions in aqueous solution.*⁵¹⁻⁵⁵ *Electrochemical hydrogenation of 1a can be accomplished by adopting a divided Co-F|NiSe two-electrode electrolyzer by using the oxidation of 1,6-hexanediamine (4a) to replace OER. Adiponitrile (5a), an important industrial feedstock for nylon production, and hydrogenated product 2a with high yields are simultaneously synthesized. To achieve a benchmark current density of 30 mA cm⁻², nearly 320 mV voltage is saved (Fig. 5c and Supplementary Note 9), showing good promise. The highly selective synthesis of tetrahydroquinoline paired with a fabrication of adiponitrile at a lower energy input shows the advantage of our method, which is otherwise difficult to access by traditional hydrogenation methods of quinoline.*”.

Second, we also do not agree with the reviewer that this manuscript is more suitable for the specialized Journal.

‘*Nature Communications*’ is a multidisciplinary journal and it covers the natural sciences, including physics, chemistry, earth sciences, medicine, biology, and all related areas. Two works on hydrogenation of quinolines separately catalyzed by a homogeneous Cobalt (Pang et al. *Nat. Commun.* 2020, 11, 1249.) and a single Iron catalysts (Long et al. *Nat. Commun.* 2020, 11, 4074.) have been recently published in ‘*Nature Communications*’. We have also recently reported a work “*Converting copper sulfide to copper with surface sulfur for electrocatalytic alkyne semihydrogenation with water*, *Nat. Commun.* 2021, 12, 3881.” in ‘*Nature Communications*’, which is deemed as one of the cutting edge research published in the field of “*Catalysis*” and highlight by the Editor (<https://www.nature.com/collections/ihbfhbiibg>).

In fact, all the other three reviewers (Reviewers 2 to 4) gave very positive comments and provided professional revision suggestions on our work. We have carefully revised the manuscript and fully addressed all the concerns/comments of the three reviewers.

In summary, our work involves material design and synthesis, organic catalysis and water electrolysis, which will be of great interest to those in the fields of materials, pharmaceutical synthesis, catalysis and energy. We believe that our manuscript is the original research of unusual urgency and significance that appeals to a broad and general audience. Thus, this work deserves being considered in ‘*Nature Communications*’ rather than a more specialized journal.

We highly appreciate the reviewer’s thorough reading and comments/suggestions about our manuscript!

To reviewer 2:

Reviewer letter: **Reviewer letter:** Combing electrocatalytic HER with hydrogenation reaction to perform organic synthesis is significant and of apparent interest. The work by Zhang and Liu reports electrocatalytic transfer hydrogenation of quinolines with H₂O using fluorine-modified Co-F nanowires as the cathode. This system exhibits high reaction selectivity in the catalytic reduction of a wide range of quinoline derivatives to THQs. To confirm that water serves as the dihydrogen source, D₂O was used for the electrochemical reduction of 2a-2d, and the corresponding deuterated THQ isotopomers were obtained in excellent yields. DFT studies and electrochemical analysis suggest that coating the Co catalyst with F enhances the adsorption of the substrate at the electrode surface for the hydrogenation, which is in a 1,4/2,3-reducing sequence via H atom transfer. The scalable electrochemical synthesis of 2a and 2b by the present method is impressive and represents a promising application. Given the importance and novelty, I recommend this work to publish in Nature Communications once the following issues are adequately addressed.

Answer: We highly appreciate the reviewer for the positive comments on our manuscript. As for the concerns or comments of the reviewer, we have provided a point-by-point response. To save the reviewer's valuable time, key revisions are displayed in a yellow background in the revised manuscript and Supplementary Materials. We are sure that the quality of this work will be greatly improved after being revised.

Comment 1: They need to rewrite the introduction and incorporate the first paragraph of "Consideration Factors" (page 4, line 67 to line 77) into the introduction part.

Answer: Thanks for the reviewer's kind suggestions. We have moved the first paragraph of "Consideration Factors" to the third paragraph of the revised introduction. Furthermore, the previous caption of "Consideration Factors and Calculation Prediction" has also been revised to "Calculation prediction for designing a Co-F electrocatalyst" in the revised manuscript.

Comment 2: In the abstract, "using a safe and clean hydrogen donor over a cost-effective catalyst" is a confusing expression. Does it mean "hydrogen donor" vs. "cost-effective catalyst"? The whole sentence should be reorganized.

Answer: Thanks for the reviewer pointing out this. We have revised the previous expression to a more appropriate form in the revised manuscript for better understanding. The revision is also extracted as follow: "using a safe and clean hydrogen donor catalyzed by a cost-effective catalyst".

Comment 3: Does "active atomic hydrogen (H*)" mean "H atom" or "H radical"? It should be appropriately defined.

Answer: Thanks for the reviewer's comments. "active atomic hydrogen (H*)" also refers to "H atom" or "H free radical" in this work, as often described in the reported literatures (*J. Am. Chem. Soc.* 2020, 142, 7036; *CCS Chem.* 2021, 3, 2669.). We have defined "active atomic hydrogen (H*)" as "active hydrogen atom" in the revised manuscript.

Comment 4: Page 2, change "promising potential" to "promising application".

Answer: We acknowledge the reviewer's kind suggestion. We have changed "promising potential" to "promising application" in the revised manuscript.

Comment 5: Page 3, correct "hydrogenating method" to "hydrogenation method".

Answer: Thanks for the reviewer's suggestion. The "hydrogenating method" has been replaced by "hydrogenation method" in the revised manuscript.

Comment 6: "Furthermore, phenyl ring is more prone to lose rather than to obtain an electron" can be simplified to "phenyl ring is prone to reduction before...".

Answer: We acknowledge the reviewer's kind suggestion. The sentence "Furthermore, phenyl ring is more prone to lose rather than to obtain an electron" has been simplified to "phenyl ring is prone to be reduced before".

Comment 7: page 5, "Our data are correctly simulated using Co-Co and Co-O two paths", how did they know their simulation is correct?

Answer: Thanks for the reviewer's comments. The Extended X-ray Absorption Fine Structure (EXAFS) is a

widely used and valuable tool to determine the structure feature and the coordination environment of an atom. Importantly, EXAFS fitting can provide more structure informations of samples, such as neighboring atoms, coordination numbers and bond distance and so on.

From the EXAFS results, two peaks located at 2.1 Å and 1.6 Å were observed in our sample, which should be corresponded to the first-shell Co–Co interaction and Co–O interaction (*ACS Energy Lett.* 2017, 2, 2545.), respectively. The Co–Co interaction of Co foil was also found at 2.1 Å, rationalizing our experimental result, which suggested a low order in the material. To better investigate the local coordination environment of our Co–F sample, we chose the Co–Co and Co–O two paths to fit the EXAFS data. According to the XAS fitting principle, R-factor is an important reference to evaluate the quality of fitting results. When the value of R-factor is less than 0.02, it indicates the reliability of the result. The fitting results revealed a R-factor value of 0.017, demonstrating the reasonability on applying of Co–Co and Co–O paths for data fitting. Furthermore, to avoid confuse to the reviewers and readers, we deleted the description of “correctly” in the revised manuscript.

Comment 8: Descriptions for some experimental details are necessary. I suggest they add the details about how the yields and selectivity were determined in the Supporting Information.

Answer: Thanks for the reviewer’s suggestions. “Conversion (Conv.)” and “Selectivity (Sel.)” are frequently used in catalytic publications (*Nat. Catal.* 2020, 3, 135; *J. Am. Chem. Soc.* 2012, 134, 17592.). In this manuscript, the Conv. of quinoline substrate and the Sel. of tetrahydroquinoline product were applied to evaluate the performance of electrocatalytic hydrogenation reaction. Gas chromatography (GC) is an useful tool for the separation and analysis of various compounds during the organic reactions. According to the GC chromatograms, the identification of products can be confirmed by the comparison of their GC retention time, and the conversion yield and product selectivity can be determined by comparing their peak area. Commonly, calibration curve of standard compounds with known concentration is usually used to determine the concentration of sample (*Nat. Catal.* 2021. <https://doi.org/10.1038/s41929-021-00721-y>; *ACS Catal.* 2018, 8, 8396.). So, we use the standard calibration curves to calculate the conversion and product selectivity of the hydrogenation reaction. The example of standard calibration curves for substrate quinoline and product tetrahydroquinoline are shown in Figure R1, which is also added to the revised Supplementary Materials (Supplementary Fig. 9).

Fig. R1 Standard calibration curves: **a** substrate, **b** product.

To be more specific, “Conv.” is calculated via dividing the amount (millimoles) of consumed substrate by originally added millimoles, and “Sel.” is calculated via dividing the obtained millimoles of tetrahydroquinoline product by the consumed millimoles of quinoline substrate. In addition, according to Reviewer 3’s suggestion, we reported the isolated yields of the products in the revised Table 1 and Fig. 5a replacing the selectivity, which are calculated via dividing the obtained millimoles of the hydrogenated *N*-heterocycle products by the theoretically formed millimoles. The equations to calculate “Conv.,” “Sel.,” and “Yield” are also found as follows:

$$\text{Conv. (\%)} \text{ (for quinoline substrate)}: = \frac{n(\text{consumed quinoline substrate})}{n(\text{initial quinoline substrate})} \times 100\%$$

$$\text{Sel. (\%)}: = \frac{n(\text{obtained tetrahydroquinoline product})}{n(\text{consumed quinoline substrate})} \times 100\%$$

$$\text{Yield. (\%)}: = \frac{n(\text{obtained tetrahydroquinoline product})}{n(\text{theoretically formed tetrahydroquinoline product})} \times 100\%$$

Moreover, we have added the detailed descriptions to Supplementary Note 4 of the revised Supplementary Materials and the related equations to Method section of the revised manuscript.

Comment 9: ²H NMR spectra of 3a, 3b, 3c must be provided to compare with the corresponding ¹H NMR spectra.

Answer: We deeply acknowledge the reviewer's wise comments. We agree that ^2H NMR test is an effective method to characterize the deuterium-containing compounds. However, due to the severe situation caused by the Coronavirus and the restricted ^2H NMR resources of our campus, we were unable to provide such characterizations. More importantly, for most of the reported papers involving the deuterated compounds, the combined use of ^1H NMR and HRMS is enough for identifying the structures and determining the deuterated ratios (e.g., *Nature* 2020, 581, 288; *Science* 2017, 358, 1182; *Nat. Catal.* 2019, 2, 1071; *Chem* 2019, 5, 2484; *J. Am. Chem. Soc.* 2019, 141, 14570; *J. Am. Chem. Soc.* 2018, 140, 10970; *J. Am. Chem. Soc.* 2012, 134, 12239; *Angew. Chem. Int. Ed.* 2019, 58, 4891; *Angew. Chem. Int. Ed.* 2019, 131, 318; *Angew. Chem. Int. Ed.* 2017, 56, 7808; *Angew. Chem. Int. Ed.* 2014, 53, 734; *Angew. Chem. Int. Ed.* 2014, 53, 230.). Few papers have used the ^2H NMR spectroscopic characterization. We have provided the corresponding NMR and HRMS data for validating the synthesized deuterated products. Therefore, the additional ^2H NMR spectroscopic characterization doesn't seem so necessary.

Comment 10: They claimed that "the Conv. of 1a is positively related to the pH value, and 1.0 M KOH gives the best result". However, the pH values for 1.0 M KOH, 0.5 M K_2CO_3 , and 0.5 M Na_2SO_4 were not given.

Answer: Thanks for the reviewer's comments. We have added the pH value of each electrolyte to the revised manuscript and Supplementary Materials (Supplementary Table 2). Additionally, according to Reviewer 3's suggestion, we have also changed 0.5 M Na_2SO_4 to 0.5 M K_2HPO_4 solution to rule out the influence of cation on the reaction. The pH values can also be found as follows: 1.0 M KOH (pH = 13.60), 0.5 M K_2CO_3 (pH = 12.33), and 0.5 M K_2HPO_4 (pH = 9.18).

In addition, to better dissolve the organic substrates, we used a mixed solvent of electrolyte and dioxane (6:1 v/v, 7 mL) in our reaction system. The actual pH values are also provided: 1.0 M KOH/dioxane (pH = 14.05), 0.5 M K_2CO_3 /dioxane (pH = 12.83), and 0.5 M K_2HPO_4 /dioxane (pH = 9.55). Slight changes of the pH values were observed when comparing with those without dioxane. We have added these data to the revised Supplementary Materials (Supplementary Table 2).

Comment 11: For "the paired synthesis of THQ and adiponitrile using a Co-F|NiSe two-electrode electrolyzer", does 1,6-hexanediamine serve as the hydrogen source in the hydrogenation of 2a?

Answer: Thanks for the reviewer's wise comments. Organic amines are often used hydrogen sources in

Department of Chemistry
Tianjin University
Tianjin 300072, P. R. China
Tel&Fax: 86-22-27403475
E-mail: bzhang@tju.edu.cn

electrochemical transformation reactions. In our work, a Nafion 117 proton exchange membrane (PEM) was used to separate the anodic and cathodic cell. Although the proton generated via 1,6-hexanediamine electrooxidation can move pass through the PEM to the cathodic cell, the hydrogenation of quinoline can be proceed easily because water are enough in the cathodic chamber.

In fact, we provide a solid proof for using water as the hydrogen source for the hydrogenation of quinoline derivatives over Co-F in a three-electrode system. Here, we only construct a two electrode system to show the potential application of paired synthesis.

We highly appreciate the reviewer's thorough reading and constructive comments/suggestions about our manuscript!

To reviewer 3:

Reviewer letter: **Reviewer letter:** This manuscript describes the preparation of a surface-fluorine-modified cobalt (Co-F) nanowire catalyst via an electroreduction process, and its application for the electrochemical transfer hydrogenation of quinolines to tetrahydroquinolines (THQ) by using H₂O as the hydrogen source. The catalyst displays good recyclability and its performance has been tested for the reduction of different *N*-heteroarenes, including the synthesis of deuterated analogues. Moreover, the synthetic value of this methodology has been further demonstrated by synthesizing bioactive precursors in a preparative scale, as well as through a paired synthesis of THQ and adiponitrile, this latter by electrochemical oxidation of 1,6-hexanediamine. With regard to the catalyst characterization, the wide range of techniques used, such as SEM, (HR)TEM, EDS, XRD, XPS, Raman XANES, and XFAFS, have revealed that the electroreduced catalyst Co-F consists on a nanowire composed by a non-reduced Co(OH)₂ core coated by fluorine-adsorbed amorphous metallic cobalt nanosheets. Particular emphasis has been placed on the elucidation of the role of the fluorine-modified surface on the electroreduction process. Based on the novelty of the synthetic strategy, the “a priori” impressive catalytic results, and the conceptual advance presented in this work, it might be suitable for Nature Communications. However, in my opinion, this version is far to be acceptable for such a world-class journal.

Answer: We highly appreciate the reviewer for the positive comments on our manuscript. As for the concerns or comments of the reviewer, we have provided a point-by-point response. To save the reviewer’s valuable time, key revisions are displayed in a yellow background in the revised manuscript and Supplementary Materials. We are sure that the quality of this work will be greatly improved after being revised.

Comment 1: Nowadays, the hydrogenation of quinolines to THQ catalyzed by non-noble metal-based catalysts is well known. However, the sentence “Additionally, noble metal and high temperature are always required to activate the hydrogen sources and/or quinoline substrates,^{5,9} which increase the possibility of undesirable hydrogenation of phenyl ring.¹²⁻¹³” seems to express the opposite. Representative works, such as the ones presented in Supplementary Table 2, and other important ones (DOI: 10.1039/C8SC02744G; DOI: 10.1021/acscatal.7b04260) should be mentioned in the introduction. In addition, reference 12 is not correctly used, since this does not reflect the over-hydrogenation of quinolines.

Answer: Thanks for the reviewer’s comments. We have rewritten the corresponding descriptions and cited the related references. In addition, we have also mentioned and cited those two representative works

(DOI: 10.1039/C8SC02744G; DOI: 10.1021/acscatal.7b04260) suggested by the reviewer in the revised manuscript. The revisions are also extracted as follows:

“Tetrahydroquinoline (THQ) skeletons are prevailing structural motifs that commonly reside in bioactive entities and natural products.¹⁻² Currently, transition metal-catalyzed hydrogenation of quinolines provides a convenient and reliable route to produce the THQ nucleus.³⁻⁶ For example, the Beller group fabricated a robust heterogeneous Fe-catalyst for the selective hydrogenation of quinolines and (iso)quinolones using molecular hydrogen as a hydrogen source.¹³ Corma and co-workers made an advance on realizing highly chemo- and regioselective hydrogenation of quinoline derivatives over a nanolayered Co-Mo-S catalyst at 110-150 °C.¹⁴ Despite impressive achievements, the dominant thermo-catalytic methods often require high-pressure H₂ and high temperature (usually ≥100 °C) to obtain satisfactory reaction efficiency. These usually not only need special apparatus with cautious operations, but also bring about cost, product selectivity, and safety concerns. Therefore, searching for an efficient and sustainable hydrogenation method that enables the hydrogenation of quinolines with sufficient activity and high selectivity, especially by applying a cheap and easy-to-handle hydrogen donor catalyzed a cost-effective catalyst at room temperature is of great significance.”

Comment 2: No electroreduction of quinolines is presented in reference 30. Therefore, the sentence “... However, conversion yields of quinolines are low.” needs to be rewritten.

Answer: Thanks for the reviewer’s comments. We are sorry that these descriptions cause a misunderstanding. In reference 30, the Lei group reported an electrochemical arylation of electron-deficient arenes through reductive activation of the substrates. The activation of quinolines via accepting an electron from the cathode is a key step for the following reaction sequences, which is also very important in our work. Therefore, we cited this paper in our manuscript. However, we did not mean that the Lei’s work is on the electroreduction of quinolines, as the reviewer stated. For a better understanding, we have rewritten the sentence as “However, the yields of coupling products of quinoline derivatives are low.” in the revised manuscript.

Comment 3: Authors commented that carried out the optimization of different adsorption modes of quinoline on the catalyst surface, and the flat adsorption resulted to be the most favored. These

comparative theoretical results should be included in the supporting information.

Answer: Thanks for the reviewer's comments. To determine the favorable adsorption mode of quinoline on the catalyst surface, we compared its flat adsorption via C and N atoms bonding and vertical adsorption via only N atom bonding on the Co(111) and Co(111)-F surfaces. A much higher adsorption energy is observed when quinoline adsorbs horizontally on the surfaces of Co based materials, demonstrating flat adsorption is a more stable adsorption configuration than vertical adsorption (Fig. R2). We have added the comparative theoretical results into the revised Supplementary Materials (Supplementary Fig. 1).

Fig. R2 Theoretical adsorption configurations of quinoline on the Co(111) and Co(111)-F surface with flat (a and b) and vertical adsorption (c and d) modes.

Comment 4: The theoretical results predict that the presence of fluorine atoms on the catalyst surface, significantly enhances the adsorption of quinoline, and boosts the electrochemical water reduction. Both should be corroborated by experimental results. However, whereas the latter prediction has been demonstrated by complementary catalytic studies with fluorine-free related materials, no experimental evidences on the different adsorption strength are reported.

Answer: Thanks very much for the reviewer's constructive comments. First, Ultraviolet-visible (UV-Vis) spectroscopy is a quantitative technique used to measure how much a chemical substance absorbs light.

We have tried to use the UV-Vis spectroscopy to detect the changes of quinoline concentration in the presence of Co foil and our Co-F material with similar electrochemical surface area. When Co foil and Co-F were immersed in different concentrations of quinoline substrate, no any changes of quinoline concentrations for all the experiments were observed. This might be beyond the detection limit by the UV-Vis spectroscopy technique.

In addition, temperature programmed desorption (TPD) is the method of observing desorbed molecules from a surface when the surface temperature is increased. Recently, Corma and co-workers have successfully observed the differences in the adsorption strength of quinoline on Co@C and tungsten doped CoW@C-0.05 surfaces by using TPD test (*ACS Catal.* 2021, 11, 8197.). Our group have also used the TPD technique to validate the adsorption differences of benzylamine on the surfaces of Pd and Pt nanoparticles (*ACS Catal.* 2021, 11, 6656.). Although we have tried to observe the desorption differences of quinoline on Co foil and Co-F, respectively, we observed no difference in desorption temperature of quinoline on both samples. We speculate that there may form a thin layer of cobalt oxide on the surface of in situ formed Co-F via electroreduction of Co(OH)F precursors during transferring it from the electrochemical cell for the TPD test because low-coordinated Co is very easy to be oxidized when exposed to air.

Furthermore, in the reported work involving catalysis, the changes of adsorption energy calculated by theory theoretical calculations are often used to predict or explain the performance origin of the materials (*Chem* 2020, 6, 2994; *Nat. Catal.* 2020, 3, 478; *Nat. Commun.* 2019, 10, 892; *Nat. Commun.* 2021, 12, 3881.). However, few reports have provided experimental evidences on how to validate the differences of adsorption strength, especially complex molecules because of the great difficulty in experimental observation. Thus, theoretical calculation can provide an alternative explanation for fundamental understanding the adsorption differences. We strongly think that the reviewer's suggestion is significant to the field of catalysis, which should raise a great attention. So, in the revised manuscript, we add the related description and cited the related references. The revised sentence is extracted as follows:

*"We have also investigated the interaction strengths of **1a** with Co and Co-F using the temperature-programmed desorption (TPD) technique as reported in the literature.³⁸ However, we observed no difference in desorption temperature of **1a** on both samples. We speculate that there may form a thin layer of cobalt oxide on the Co-F surface during transferring it from the electrochemical cell for the TPD test."*

Comment 5: To avoid readers' confusion, graphics a) and b) in Figure 1 should be represented in the same

adsorption energy scale.

Answer: Thanks for the reviewer's kind suggestions. We have renewed Figs. 1a and 1b by using the same adsorption energy scale of the Y-axis. These revised figures are also extracted as follows (Fig. R3):

Fig. R3 a and b Comparisons of E_{ads} of **1a** and **2a** on pure Co and Co-F, respectively (insert: the stable adsorption modes of **1a** and **2a**).

Comment 6: According to the authors' statement, "The exposing metallic Co works as the active center for the electrochemical transfer hydrogenation reaction of quinolines." To confirm this hypothesis, the electrocatalytic reaction in the presence on the non-activated Co(OH)F should be tested. If this is the case, since similar cathodic potential is used (around -1.2 V) for both processes, i.e. catalyst activation and catalytic reaction, an induction period in the formation of THQ should be observed.

Answer: We acknowledge the reviewer's valuable suggestions. We have carried out time-dependent experiments of electrochemical hydrogenation of quinoline **1a** over non-activated Co(OH)F and electroreduced Co-F cathode within 3 h, respectively. The conversion of **1a** can reach to about 20% using Co-F as the cathode within the initial 0.5 h, whereas no obvious conversion of **1a** is observed (Fig. R4). After about 1.5 h, the conversion of **1a** increases obviously, similar with those over the Co-F cathode within the first 2 h. The induction period of Co(OH)F under electroreduction conditions might hint that the electroreduced Co-F works as the active center for the electrochemical hydrogenation reaction of quinolines. We have also included these results and the corresponding descriptions to the revised manuscript and Supplementary Materials (Supplementary Fig. 11). The descriptions are also extracted as follows:

“Furthermore, we observe an obvious induction period within about the initial 0.5 h when using Co(OH)F precursor as the cathode under similar reaction conditions (Supplementary Fig. 11), which may further support metallic Co serving as the active center for this reaction.”

Fig. R4 Conv. comparisons of electrochemical hydrogenation of **1a** over Co-F and Co(OH)F within 3 h.

Comment 7: The term “conversion yield”, used along the manuscript, leads to confusion. Typically, “conversion” is referred to the substrate (i.e. quinoline), while the term “yield” is used for products.

Answer: We do agree and thanks for the reviewer’s suggestions. In the revised manuscript, we have changed “conversion yield” to “conversion” for the substrate and “yield” for the products.

To be more specific, “Conv.” is calculated via dividing the amount (millimoles) of consumed substrate by originally added millimoles, and “Sel.” is calculated via dividing the obtained millimoles of tetrahydroquinoline product by the consumed millimoles of quinoline substrate. In addition, according to Reviewer 3’s suggestion, we reported the isolated yields of the products in the revised Table 1 and Fig. 5a replacing the selectivity, which are calculated via dividing the obtained millimoles of the hydrogenated *N*-heterocycle products by the theoretically formed millimoles. The equations to calculate “Conv.”, “Sel.”, and “Yield” are also found as follows:

$$\text{Conv. (\%)} \text{ (for quinoline substrate)}: = \frac{n(\text{consumed quinoline substrate})}{n(\text{initial quinoline substrate})} \times 100\%$$

$$\text{Sel. (\%)}: = \frac{n(\text{obtained tetrahydroquinoline product})}{n(\text{consumed quinoline substrate})} \times 100\%$$

$$\text{Yield (\%)} = \frac{n(\text{obtained tetrahydroquinoline product})}{n(\text{theoretically formed tetrahydroquinoline product})} \times 100\%$$

Moreover, we have added the detailed descriptions to Supplementary Note 4 of the revised Supplementary Materials and the related equations to Method section of the revised manuscript.

Comment 8: There is no clear information on the quantification methods used to determine conversion and selectivity. From the equations shown in the methods section, it seems both have been calculated by GC. However, no internal standard were added to the reaction mixture. Actually, it is a highly unreliable method because side products are easily overlooked and underestimated. Moreover, comments on Figure 2b mention that selectivity (99%) is determined according to Supplementary Note 3, which refers to GC-MS measurements, being this a qualitative rather than a quantitative tool.

Answer: Thanks for the reviewer's comments. Conversion is referred to the substrate, which is calculated via dividing the amount (millimoles) of consumed substrate by originally added millimoles.

The consumed amount of substrate is quantified according to GC measurements using the standard calibration curve, which is another commonly used and reliable method for quantifying the substance (*Nat. Catal.* 2021. <https://doi.org/10.1038/s41929-021-00721-y>; *ACS Catal.* 2018, 8, 8396.). So, we use the standard calibration curves to calculate the conversion yield and product selectivity of the hydrogenation reaction. The example of standard calibration curves for substrate quinoline and product tetrahydroquinoline are shown as follow, which can also be seen from the reply to reviewer 2.

Fig. R1 Standard calibration curves: **a** substrate, **b** product.

In addition, we do agree with the reviewer that GC-MS measurement is a qualitative rather than a

quantitative tool and we deeply acknowledge the reviewer for pointing out the mistake on using GC-MS for selectivity calculation. In fact, we use GC measurements to quantify the selectivity and we have revised the clerical error in the revised Supplementary Note 3. Furthermore, to better understand, we provide the isolated yield for each product in the revised Table 1 and Fig. 5a.

Comment 9: The F 1s XPS spectrum of the reused catalyst shown in Supplementary Fig. 8 does not reflect the loss of surface fluorine species. For that, the surface atomic ratio Co/F should be calculated and compared with that of the fresh catalyst (i.e. with that one just obtained after the electrocatalytic activation). Could be possible to reactivate the reused catalyst after eight runs?

Answer: We thank the reviewer's comments. We calculate the surface F content of the reused catalyst by the quantitative analysis of X-ray photoelectron spectroscopy (XPS) spectroscopy. The F content is lower than of the fresh catalyst (Fig. R5), reflecting the losing of F after recycling experiment. We have incorporated this to the revised Supplementary Fig. 12.

Fig. R5 Comparisons of surface F content on the fresh and reused Co-F catalyst.

According to the reviewer's suggestion, we add a small amount of KF (5mg, 0.017 M) in to the reaction system for the ninth run. However, the conversion of **1a** decreases further rather than increasing during the same reaction time (Fig. R6). We speculate that the decreased activity of Co-F after eight runs may be mainly ascribed to the deactivation of Co catalytic sites caused by the quinoline substrate or hydrogenated product. In addition, deactivation of metal catalysts is indeed a common but tricky problem

in metal-catalyzed hydrogenation of quinolines in reported literatures (*Angew. Chem. Int. Ed.* 2020, 59, 17408; *ACS Catal.* 2016, 6, 5816.). Therefore, we revised our previous descriptions on stability test. The revisions are extracted as follows:

“Moreover, to evaluate the stability of Co-F, this electrode is repeatedly used for the next electrochemical experiment by adding a same amount of **1a** after the simple washing with ethanol and DI water. Fig. 2d shows no obvious decreases in the conversion of **1a** and selectivity of **2a** in 7 runs at -1.1 V vs. Hg/HgO and within the same reaction time, demonstrating a good stability of Co-F electrode. However, for the eighth run, the decreased conversion of **1a** may be mainly ascribed to the losing of surface F (Supplementary Fig. 12) and the deactivation of Co catalytic sites caused by the substrate or product.”

Fig. R6 Stability tests of Co-F cathode for electrochemical hydrogenation of **1a**.

Comment 10: In the mechanism studies, authors state that “... we observe a lower onset potential and a significantly enhanced Conv. of **1a** after adding NaF into the electrolyte over a Co foil cathode (Figs. 3a-b).” On the contrary, Fig. 3a shows the opposite. These results seem to conflict with each other, why?

Answer: We acknowledge the reviewer for pointing out this. In fact, our Co-F cathode shows enhanced an performance toward hydrogenation reaction (HER) compared with that of Co foil. The blue and red LSV curves are mislabeled in previous Fig. 3a. We have renewed it in the revised manuscript, which can also be found from Fig. R7.

Fig. R7 LSV curves recorded for HER at a scan rate of 10 mV s^{-1} over Co foil with and without NaF in a mixed solvent of 1.0 M KOH with dioxane.

Comment 11: Authors carried out a study depending on the pH by using different electrolytes, such as KOH, K_2CO_3 , and Na_2SO_4 . However, they previously remark the importance of the cation in the formation of a specific anion-hydrated cation pair, which seems to be crucial for accelerating H_2O activation. Therefore, to rule out the effect of the cation in the pH variation study, the same cation should be used for all experiments. That is, Na_2SO_4 needs to be substituted by K_2SO_4 . On the other hand, pH values should be included in Supplementary Figure 11a.

Answer: Thanks very much for the reviewer's constructive suggestions. We have tried to use K_2SO_4 (0.5 M) to replace Na_2SO_4 to rule out the effect of the cation on the reaction. However, it is failed due to the precipitation of K_2SO_4 from the mixed solution even if we adding a small amount of dioxane (to better dissolve quinoline substrate). Therefore, we use 0.5 M K_2HPO_4 in the pH variation studies.

Table R1 pH values of different electrolytes (6.0 mL) with and without adding dioxane (1.0 mL).

	without dioxane	with dioxane
1.0 M KOH	13.60	14.05
0.5 M K_2CO_3	12.33	12.83
0.5 M K_2HPO_4	9.18	9.55

In addition, in our electrochemical experiments, we used a mixed solvent of electrolyte and dioxane (6:1 v/v, 7 mL) to better dissolve quinoline substrate. The pH values of the solvents with and without adding dioxane are measured using FiveEasy Plus™ pH meter with pH electrode LE438 (the Mettler Toledo Company), as depicted in Table R1. Slight changes of the pH values were observed after adding dioxane. The results of pH variation studies reveal that the Conv. of quinoline **1a** is positively related to the pH value, and 1.0 M KOH gives the best result. We have added these data to the revised Supplementary Materials (Supplementary Table 2).

Comment 12: Authors should explain in more detail why they think that the use of the quinoline derivatives **1q**, **1r**, and **1s** are good candidates to discern between the different adsorption modes of quinolines.

Answer: Thanks for the reviewer's comments. According to our theoretical calculations, flat adsorption of quinoline substrate is the more favorable adsorption configuration on the Co (111) surface. The favorable flat adsorption of quinoline-based substrate over a heterogeneous catalyst surface is also theoretically validated in other reported literatures for thermocatalysis (*Chem* 2020, 6, 2994; *Nat. Commun.* 2020, 11, 4074.). In our control experiments, the quinoline derivatives **1q**, **1r**, and **1s** are all sterically hindered substrates with the substituents at the C₂, C₈, or N positions. Such steric effect may exert significant influences on the electrochemical hydrogenation reaction if the substrate adsorbs in other modes rather than the flat adsorption, or adsorption only via N atom bonding. Acceptable (**1q** and **1r**) and excellent (**1s**) conversion yields are observed under the similar reaction conditions, which may provide an indirect support on the flat adsorption of quinoline in our reaction. Furthermore, using the steric effect of substrate to validate the proposed mechanism are often seen in literature (*Sci. Adv.* 2020, 6, eabb3831.).

To describe accurately, we revise the related description "*These results may hint flat adsorption of 1a on the Co-F surface*" to "*These results may provide an indirect support on the flat adsorption of quinoline in our reaction*" in the revised manuscript.

Comment 13: In Table 1, isolated yields of the obtained products should be included. In addition, it would be interesting to further investigate the chemoselectivity of this electrochemical reduction process by testing N-heterocycles containing other reducible functional groups, such as double and triple C-C bonds,

ketone, aldehyde, nitrile, ester, and amide groups.

Answer: Thanks very much for the reviewer's comments. In the revised Table 1 and Fig. 5a, we have provided isolated yields of the obtained products for ease of understanding

Furthermore, according to the reviewer's kind suggestions, we have also examined *N*-heterocycles containing other reducible functional groups. The amide group is compatible under our reaction conditions, giving rise to the corresponding product with 80% isolated yield. However, for quinoline substrates containing the $-C\equiv CH$, $-CN$, and $-CHO$ groups, those fragile groups are hardly to survive and a mixture of the hydrogenated products are obtained. We use GC-MS for a qualitative analysis of possible products for each substrate, as seen in Figs. R8-10. However, due to small amount of each products and with very close polarity in each reaction system, we can't purify each product for an accurate isolated yield with NMR tests. Therefore, precise screening of reaction conditions or further modification of the electrocatalyst to improve the compatibility of such more readily reducible functional groups will be highly needed in the future work. These data and corresponding descriptions have also been included into the revised manuscript and Supplementary Materials (Supplementary Figs. 18-20). The descriptions can also be extracted as follow: *"To further investigate the chemoselectivity of this electrochemical hydrogenation reaction, we have also examined the quinoline substrates containing readily reducible functional groups, such as $-C\equiv CH$, $-CN$ and $-CHO$, under our reaction conditions. Unfortunately, those fragile groups are hardly to survive and a mixture of the hydrogenated products are obtained (Supplementary Figs. 18-20). Therefore, precise screening of reaction conditions or further modification of the electrocatalyst to improve the compatibility of such more readily reducible functional groups will be highly needed in the future work."*

Fig. R8 A qualitative analysis of possible products by GC-MS for 6-ethynylquinoline substrate.

Fig. R9 A qualitative analysis of possible products by GC-MS for quinoline-6-carbonitrile substrate.

Fig. R10. A qualitative analysis of possible products by GC-MS for quinoline-6-carbaldehyde substrate.

Comment 14: In general, for the convenience of the readers, the reactions conditions used in each of the experiments shown in Figures should be indicated in the Figure caption.

Answer: Thanks very much for the reviewer's valuable suggestion. We have added the main reaction conditions of each experiment to the Figure captions in the revised manuscript.

We highly appreciate the reviewer's thorough reading and constructive comments/suggestions about our manuscript!

To reviewer 4:

Reviewer letter: Summary: Fluorine-modified cobalt (Co-F) nanowires were synthesized and used for the electrochemical hydrogenation of quinolines. The cobalt with metallic character and fluorine present was found to both be key to the successful hydrogenation of the quinolines. In alkaline conditions of KOH, the reaction proceeded without limitations, able to achieve 100% conversions and 99-100% selectivities. No unsaturated ring hydrogenation was observed. Electrochemical hydrogenation is advantageous for selective hydrogenation, particularly when ring saturation is undesired. The catalyst proposed in the work holds significant promise for its selectivity and activity. The work could be enhanced through how it is presented and changing at what point experiments are compared to each other.

Answer: We highly appreciate the reviewer for the positive comments on our manuscript. As for the concerns or comments of the reviewer, we have provided a point-by-point response. To save the reviewer's valuable time, key revisions are displayed in a yellow background in the revised manuscript and Supplementary Materials. We are sure that the quality of this work will be greatly improved after being revised.

Comment 1: While it is great to show that 100% conversions can be reached, illustrating that the reactions are not equilibrium limited, catalytic comparisons at or near 100% conversion have diminished value. Catalytic studies to allow for distinguishing of catalysts and conditions are better done far away from 100% conversion. This is because 100% conversion results do not distinguish how long it took to get to 100% conversion – has the system been sitting there for a few hours or just a few minutes? This is especially true in the catalyst recycle results. The catalyst is likely deactivating after each run, though only at run 7 and beyond is the deactivation significant enough to slow the reaction down to not be able to go to completion in the given time.

Answer: We sincerely thank the reviewer's valuable comments. We do agree with the reviewer that "Catalytic studies to allow ... to get to 100% conversion". First, we have carried out detailed investigations on the conversion of quinoline **1a** with the reaction time. Fig. R11 reveals that near 100% conversion of **1a** can be finished within about 6 h at -1.1 V vs. Hg/HgO. And, **1a** conversion yield increases slowly after about 2.5 h, which may be ascribed to the low concentration of **1a** causing mass transport limitation to the electrode interface. Second, in the performance comparison experiments (Fig. R11), we compared the conversion of quinoline substrate **1a** and selectivity of product **2a** over different cathodes at -1.1 V vs.

Hg/HgO within the same reaction time of 6 h. Although similar selectivity of **2a** are obtained for all the tested cathodes, the conversion yield of **1a** over our Co-F cathode is much higher than that over other cathodes, reflecting the high activity of the Co-F cathode.

Fig. R11 **a** Time-dependent **1a** conversion (Conv.) and **2a** selectivity (Sel.) over Co-F. **b** Comparisons with other Co-based electrodes.

Furthermore, to evaluate the stability of Co-F, this electrode is repeatedly used for the next electrochemical experiment by adding the same amount of **1a** after being washed several times with ethanol and deionization water. Fig. 2d shows no obvious decreases in the conversion of **1a** and selectivity of **2a** in 7 runs at -1.1 V vs. Hg/HgO and within the same reaction time, demonstrating a good stability of Co-F electrode. However, for the eighth run, the decreased conversion of **1a** may be mainly ascribed to the losing of surface F (Supplementary Fig. 12) and the deactivation of Co catalytic sites caused by the substrate or product. In addition, deactivation of metal catalysts is indeed a common but tricky problem in metal-catalyzed hydrogenation of quinolines due to the interaction of nitrogen atom of substrates or products with the supported metal centers in reported literatures (*Angew. Chem. Int. Ed.* 2020, 59, 17408; *ACS Catal.* 2016, 6, 5816.). For a better understanding, we have added the main reaction conditions to the figure captions in the revised manuscript, as reviewer 3 suggested.

Comment 2: In the discussion of pH, the actual solution pHs should be reported (for the whole solution, not just components of the solution). It is also not clear if the pH studies are actual anion effect studies. The anion can also have an impact on the electrochemical performance—perhaps the anion is replacing F—for example. This may be independent of the pH.

Answer: Thanks for the reviewer's kind comments. First, according to Reviewer 3's suggestion (**Comment 11**), we have changed 0.5 M Na₂SO₄ to 0.5 M K₂HPO₄ solution to rule out the influence of cation on the reaction. The pH values of each electrolyte are measured using FiveEasy Plus™ pH meter with pH electrode

LE438 (the Mettler Toledo Company), as seen in Table R1 for the reply to Reviewer 3.

Table R1 pH values of different electrolytes (6.0 mL) with and without adding dioxane (1.0 mL).

	without dioxane	with dioxane
1.0 M KOH	13.60	14.05
0.5 M K ₂ CO ₃	12.33	12.83
0.5 M K ₂ HPO ₄	9.18	9.55

Additionally, in our electrochemical experiments, we used a mixed solvent of electrolyte and dioxane (6:1 v/v, 7 mL) to better dissolve the organic substrates. The actual pH values of each mixed solvent are also measured using FiveEasy Plus™ pH meter. From Table R1, slight changes of the pH values are observed with and without adding dioxane. We have added these data to the revised Supplementary Materials (Supplementary Table 2).

Furthermore, we agree with the reviewer that *the anion may also have an impact on the electrochemical performance*. In our work, we use 1.0 M KOH as the electrolyte and its pH value is about 13.6, similar to the literature (*ACS Energy Lett.* 2020, 5, 1083.). We have tried electrochemical hydrogenation of **1a** with KOH at different pH values (e.g., pH = 12, 10). However, we are failed due to the low concentration of conducting ions in the solution causing very large resistance. Therefore, according to reported literatures on investigating the influence of pH on the reactions (*Nat. Catal.* 2020, 3, 478; *Nat. Commun.* 2019, 10, 892.), we select different types of electrolytes with good conductivity to examine the pH influence on our reaction.

We thank the reviewer again for raising a good suggestion on “*perhaps the anion is replacing F-*”. We will pay special attentions to it in our future work. In this work, we need focus on the F effect. The reviewer’s nice suggestions will be highly helpful for our future research.

Comment 3: pH studies should be reported vs RHE and studied at constant overpotential. As presented, the overpotentials are dramatically different from each other. Even when the same electrolyte and catalyst were used, the overpotentials can lead to different results (ex. Fig 2 a).

Answer: Thanks for the reviewer's comments. According to the reviewer's suggestion, we first carry out the Linear Sweep Voltammetry (LSV) experiments of the Co-F electrode in the electrolyte with different pH values with Hg/HgO as the reference electrode. The LSV curves in Fig. R12a reveal close overpotentials for HER, and we therefore select -0.2 V vs. RHE to investigate the pH influence on electrochemical hydrogenation of **1a**. Fig. R12b displays that the Conv. of **1a** is positively related to the pH value, and 1.0 M KOH gives the best result. These results may demonstrate that electrochemical hydrogenation of **1a** is easy to proceed at a higher pH value. The renewed data are also included into the revised manuscript and Supplementary Materials (Supplementary Fig. 15).

Fig. R12 a LSV curves of Co-F cathode at a scan rate of 10 mV s^{-1} in the mixed solvent of dioxane with different electrolytes. **b 1a** Conv. and **2a** Sel. obtained in the mixed solvent of dioxane with different electrolytes. Reaction conditions: **1a** (0.1 mmol), Co-F (working area: 1.0 cm^2), dioxane/electrolyte (1:6 v/v, 7 mL), -0.2 V vs. RHE, RT, 6 h.

Comment 4: The experiment with 1 atm H_2 is a good start at showing that the electrochemical hydrogenation is not from H_2 produced via HER that is then further used for hydrogenation. The experiment is incomplete, however. This is because the cobalt can be partially oxidized in the KOH when at open circuit potential. When a reducing potential is applied, the cobalt is reduced. This means that the catalyst being studied at open circuit potential with H_2 gas is not the same catalyst and would not have the same activity as the one being studied in electrochemical hydrogenation. A good follow-on is to do the electrochemical hydrogenation with 1 atm H_2 being sparged through the solution. Compare this to the electrochemical hydrogenation experiment without H_2 to prove that H_2 is not the reactant.

Answer: Thanks for the reviewer's kind suggestion. According to the reviewer's suggestion, in order to

exclude the influence of H₂O and the oxidation of Co-F on the hydrogenation reaction, electrochemical hydrogenation of **1a** is carried out in an anhydrous acetonitrile (CH₃CN) solution using 0.2 M tetrabutylammonium tetrafluoroborate (TBATF₄) as electrolyte with and without 1 atm H₂, respectively. After 10 h, no any hydrogenated product **2a** is detected for each reaction system (Scheme R1). This may rule out in situ formed H₂ via HER serving as potential hydrogen source for electrochemical hydrogenation of quinolines. We have added the renewed data to the revised Supplementary Materials (Supplementary Scheme 2).

Scheme R1 Control experiments on electrochemical hydrogenation of **1a** in an anhydrous CH₃CN solution using 0.2 M TBATF₄ as electrolyte in the absence and presence of 1 atm H₂.

Comment 5: Nickel foam is used as the support for the catalyst. No mention of any role of nickel is mentioned in the study and if it has any catalytic role.

Answer: Thanks for the reviewer's comments. We have tested the performance of nickel foam for electrochemical hydrogenation of **1a** at different applied potentials under other identical reaction conditions. However, nearly no any conversion of **1a** is observed (Fig. R13). This may further demonstrate that the Co-F catalyst contributes to the high activity for this electrochemical hydrogenation reaction. These results and the corresponding descriptions are also included into the revised manuscript and Supplementary Materials (Supplementary Fig. 10). The descriptions can also be extracted as follows:

“And, the nickel foam (NF) almost shows no activity toward this electrochemical hydrogenation reaction even if at different applied potentials (Supplementary Fig. 10).”

Fig. R13 Electrochemical hydrogenation of **1a** over nickel foam cathode at different applied potentials.

Comment 6: It is not clear if the electrochemical system studied is homogeneous or biphasic. Mixing is said to lead to a homogeneous solution though an aqueous layer is also formed. This is confusing and should be clear up front.

Answer: Thanks for the reviewer's comments. Dioxane is highly soluble and stable in water (<https://www.sciencedirect.com/topics/chemistry/1-4-dioxane>) and a homogeneous solution is formed when mixing 1.0 mL of dioxane and 6.0 mL of KOH in our reaction.

In addition, to determine whether electrochemical hydrogenation of **1a** occurs in the bulk solution or on the surface of Co-F electrode, we employ 1-dodecanethiol as capping reagents to modify the Co-F cathode before the electrolysis begins. The conversion of **1a** is obviously decreased (Fig. R14). This may demonstrate that electrocatalytic hydrogenation of **1a** mainly proceeds on the Co-F surface. It means that this electrochemical reaction occurs at the interface between electrode and solution rather than the dioxane/water interface. We have also added these data and the corresponding descriptions into the revised manuscript and Supplementary Materials (Supplementary Fig. 16). The descriptions are also extracted as follows:

*“First, to determine whether electrochemical hydrogenation of **1a** occurs in the bulk solution or on the surface of Co-F electrode surface, we employ 1-dodecanethiol to modify the Co-F cathode. The conversion of **1a** is obviously decreased under the standard reaction conditions after introducing 1-dodecanethiol to the reaction system (Supplementary Fig. 16). This may reveal that electrocatalytic hydrogenation of **1a** proceeds mainly on the Co-F surface.”*

Fig. R14 Electrochemical hydrogenation of **1a** over the 1-dodecanethiol modified Co-F cathode.

Comment 7: 1.0 M KOH or other molarities are actually not used in the studies. Additional solvent and reactant is added to a solution already containing 1.0 M KOH. This dilutes the base (or other electrolyte). The concentrations used in the electrochemistry, not in solutions prior to being combined should be used and reported.

Answer: We sincerely acknowledge the reviewer's kind comments. We have provided the actual concentrations of electrolyte and reactant in the revised manuscript and Supplementary Materials (Supplementary Table 2).

Comment 8: Concentrations, not moles should be reported for the quantities of reactants used. This allows the work to be much more transferrable.

Answer: We thanks the reviewer's comments. "moles" is often used to quantify the reactant in reported papers involving catalysis (*Nat. Catal.* 2020, 3, 135; *Nat. Chem.* 2019, 11, 242; *Nat. Commun.* 2021, 12, 4968; *ACS Catal.* 2018, 8, 4545.). However, we also respect the reviewer's suggestion and we use "concentration" to replace "mmol" for the quantities of reactants in the revised manuscript.

Comment 9: 'electrochemical transfer hydrogenation' is awkward and not customarily used. Rather, 'electrochemical hydrogenation' is usually the phrase used for the reactions being studied.

Answer: Thanks for the reviewer's comment. We have changed all descriptions of "*electrochemical transfer hydrogenation*" to "*electrochemical hydrogenation*" for easy understanding in the revised manuscript and Supplementary Materials.

Comment 10: The title is also awkward and lacks information. Cobalt is also key. Perhaps 'Fluorine-modified cobalt catalysts for electrochemical hydrogenation of quinolines at room temperature' or something like this would be better and more informative.

Answer: We deeply thank for the review's wise comments. We have changed previous title to a more accurate one "*Fluorine-modified cobalt electrocatalyst for hydrogenation of quinolines with water at room temperature*" in the revised manuscript.

Comment 11: Justification of doing electrochemical hydrogenation because hydrogen is a safety hazard is difficult to make because the petrochemical industry safely uses high pressure hydrogen regularly. Rather, focusing on selectivity advantages and decarbonization strategies would be more convincing.

Answer: Thanks very much for the reviewer's constructive suggestions. We have rewritten the previous descriptions to make them more convincing. The related sentences are extracted as follows:

"Despite impressive achievements, the dominant thermo-catalytic methods often require high-pressure H₂ and high temperature (usually ≥ 100 °C) to obtain satisfactory reaction efficiency. These usually not only need special apparatus with cautious operations, but also bring about cost and product selectivity concerns, and safety risks."

Comment 12: Line 72-73. Stating that Co-based cathodes are widely used in hydrogen production is misleading. Commercial HER catalysts do not involve Co. Recommend instead of widely used stating widely studied.

Answer: Thanks for the reviewer's suggestion. We have changed the previous "*widely used*" to a more reasonable "*widely studied*" in the revised manuscript.

Comment 13: Figure 1 caption. Reword to emphasize the synthesis of Co-F NWs. It is easy to miss that the Co-F was electrochemically synthesized so the caption gets confusing. Also, by having the caption clearer, the message about how Co-F was synthesized becomes clearer. ‘Synthesis of Co-F NWs by electroreduction of Co(OH)F NWs’.

Answer: Thanks very much for the reviewer’s comments. We have changed the previous caption of Fig. 1 to “*Synthesis of Co-F NWs by electroreduction of Co(OH)F NWs*” to make it more clear in the revised manuscript.

Comment 14: Fig. 2 & 3 captions. Conditions need to be stated including reaction duration times and electrolytes.

Answer: Thanks very much for the reviewer’s valuable suggestion. We have added the main reaction conditions of each experiment to the Figure captions in the revised manuscript.

Comment 15: Make sure to define “NF” in text and experimental sections. While this stands for ‘nickel foam’ it could just as easily stand for ‘nanofiber’.

Answer: We sincerely thank the review for the kind suggestion. “NF” is referred to “nickel foam”, which is defined in the revised manuscript and Supplementary Materials.

Comment 16: Line 197. Provide a reference for the non-covalent Coulomb interaction.

Answer: We acknowledge the reviewer’s comments. The related references are provided in the reference section as Refs. 46 and 47 in the revised manuscript.

Comment 17: Line 202. Define what ‘n’ is.

Answer: Thanks for the reviewer’s comment. “n” is referred to the number of ionic hydration, which is defined in the revised manuscript.

Comment 18: Line 242. 'heavily dragged' is a strange phrase.

Answer: We thank the reviewer for pointing out this. We have changed "*heavily dragged*" to "*significantly impeded*" for conveniently reading in the revised manuscript.

Comment 19: A table of contents for the supplemental information would be appreciated.

Answer: Thanks very much for the reviewer's comments. We have added a table of contents in the revised Supplementary Materials, which can also be seen as follows:

Supplementary Figure 1. Theoretical adsorption configurations of quinoline on the Co(111) and Co(111)-F surfaces.

Supplementary Figure 2. SEM, TEM, HRTEM, and elemental mapping images of Co(OH)F NWs.

Supplementary Figure 3. Synthesis of Co-F NWs by electroreduction of Co(OH)F NWs.

Supplementary Figure 4. Experimental and simulated XAFS spectra of Co-F at the Co K-edge.

Supplementary Figure 5. XRD patterns and EDS results of Co(OH)F NWs and Co-F NWs.

Supplementary Figure 6. SEM, TEM, HRTEM images and elemental mapping of Co-F NWs.

Supplementary Figure 7. Reaction setup.

Supplementary Figure 8. Performances studies and contrast experiments.

Supplementary Figure 9. Standard calibration curves for quantitative analysis of quinoline and tetrahydroquinoline.

Supplementary Figure 10. Electrochemical hydrogenation of **1a** over a nickel foam cathode at different applied potentials.

Supplementary Figure 11. Conv. comparisons of electrochemical hydrogenation of **1a** over Co-F and Co(OH)F, respectively, within 3 h.

Supplementary Figure 12. XPS spectra of the Co-F before and after stability test. **Supplementary Figure 13.** EIS plots and double-layer capacitance of Co(OH)F NWs and Co-F NWs.

Supplementary Figure 14. The effects of cations on the hydrogenation of **1a** over Co foil.

Supplementary Figure 15. The pH variation studies.

Supplementary Figure 16. Electrochemical hydrogenation of **1a** over 1-dodecanethiol modified Co-F cathode.

Supplementary Figure 17. Proposed reaction mechanism for electrochemical hydrogenation of quinolines with H₂O over a Co-F cathode.

Supplementary Figure 18. A qualitative analysis of possible products by GC-MS for 6-ethynylquinoline substrate.

Supplementary Figure 19. A qualitative analysis of possible products by GC-MS for quinoline-6-carbonitrile substrate.

Supplementary Figure 20. A qualitative analysis of possible products by GC-MS for quinoline-6-carbaldehyde substrate.

Supplementary Table 1. EXAFS fitting parameters of a Co-F catalyst.

Supplementary Table 2. pH values of different electrolytes (6.0 mL) with and without adding dioxane (1.0 mL).

Supplementary Table 3. Comparisons of the hydrogenation of quinolines to tetrahydroquinolines by representative heterogeneous catalytic methods over non-noble metal catalysts and our method.

Supplementary Scheme 1. Deuterated experiment.

Supplementary Scheme 2. Electrochemical hydrogenation of **1a** in anhydrous CH₃CN in the absence and presence of H₂ over Co-F.

Supplementary Scheme 3. Electrochemical hydrogenation of **1a** with H₂O over a Co-F NWs cathode at a constant current density of -100 mA cm⁻².

Supplementary Notes 1-9.

Supplementary Figures 21-41. NMR spectra of **2a-2q**.

GC-MS spectra of 2r-2s and 5a.

References (1-15). ”

We highly appreciate the reviewers' thorough reading and constructive comments/questions about our manuscript! We thank all kind and professional suggestions from the reviewers.

We are sure that the quality of this work will be greatly improved according to these nice comments and wise suggestions.

Reviewers' comments:

Reviewer #2 (Remarks to the Author):

This referee appreciates the authors' thoughtful responses. I think my previous concerns have been adequately addressed, so I recommend publishing this work in Nature Communications.

Reviewer #3 (Remarks to the Author):

The revised manuscript by Bin Zhang, Cuibo Liu, and co-workers represents a hugely significant improvement on the original submission. Important previously unanswered questions have been addressed and discussed in a scholarly way with new supporting data. However, I remain concern on the methodology followed to quantify the conversion, yield and selectivity by GC. The equations used are correct, however, the problem arises from the fact that no internal standard was used. Authors based their calculations on standard calibration curves in which they represent peak area versus concentration. Although this method seems to be previously applied elsewhere, avoiding the use of internal (or external) standard is not a good practice of quantification in catalysis, especially when this is carried out in a GC-FID equipment. Actually, a correct calibration curve have to be constructed by representing: (molar ratio of substrate to the standard) versus (area ratio of substrate to the standard). Unfortunately, this means that authors have not properly worked, and consequently, all GC-based values reported on the manuscript should be revised before the manuscript being accepted, which, that to the reviewer's opinion, only should be occur if results do not differ too much from the previous ones. Furthermore, the following issues should also be addressed:

- Comments on products 2a, 2b, 2c, 2d, 2f should be included in the manuscript.
- The chemical structure of quinoline-6-carbonitrile and 1,2,3,4-tetrahydroquinoline-6-carbonitrile are wrongly drawn in Supplementary Figure 19.
- Authors state in the response to referees' letter that "The amide group is compatible under our reaction conditions, giving rise to the corresponding product with 80% isolated yield." However, a 90% isolated yield can be found in Table 1 of the manuscript.
- The meaning of the revised sentence "Furthermore, the phenyl ring is prone to reduction before initiating desired reaction sequences under electro- or photocatalytic reaction conditions" is different from the sentence of the former manuscript "Furthermore, phenyl ring is more prone to lose rather than to obtain an electron before initiating desired reaction sequences under electro- or photo-catalytic reaction conditions.". Please, revise it!!

Reviewer #4 (Remarks to the Author):

Remarks for revision:

- There are still significant work that would need to be done before this manuscript could be considered

for publication.

- It is still not clear to the reviewer why the work of the manuscript is so significant that it be worthy of Nature Communications instead of an organic chemistry or electrochemistry-based journal.
- Throughout the manuscript H₂ and water electrolysis are mentioned as positives for the work. In electrochemical hydrogenation, hydrogen generation is seen as a loss of electrons to undesired reactions, not a favorable occurrence. It is true that water activation is necessary for both electrochemical hydrogenation and water electrolysis. It is undesired to do the full reduction to H₂, however. This needs to be clarified and corrected throughout the manuscript, abstract and captions.
- The revised justification of the work states that the reason to do electrochemical hydrogenation rather than traditional, well-developed thermocatalytic methods is because H₂ gas safety concerns and high temperatures of 100-150C. This does not motivate the chemical industry who would find 100-150C as low temperature and frequently works with hydrogen gas safely. Motivating factors to consider are decarbonization through use of renewable electricity and no need of hydrogen produced from methane steam reforming (a lot of CO₂ produced in H₂ generation), or distributed manufacturing where hydrogen as a utility is not readily available.
- Lines 64-67. '...the phenyl ring is prone to reduction before initiating desired reaction sequences under electro- or photocatalytic reaction conditions. Therefore, electrochemical hydrogenation will avoid the hydrogenation of phenyl rings of quinolines and improve regioselectivity.' These two sentences conflict with each other.
- Figure 2d and discussion around recyclability of the catalyst. The catalyst is likely decaying during each run, though only cycles 8 and 9, did the decay become significant enough to notice at 6 hours. Conversion as a function of time for each of the cycles would be more useful in really seeing if the catalyst is stable. Alternatively, plotting at 3 hours where the reaction is not complete on fresh catalyst will allow the differentiation of the catalyst as it ages. Right now, the results presented in figure 2d are misleading.
- Line 165. Catalyst is stated as being quickly used after synthesis. Is this indicating the catalyst is unstable?
- Line 177. The catalysts in 2c are not acknowledged, only those in the SI Fig 8c are acknowledged.
- Line 203. The time at the measurement is being compared, not just the conversion should be indicated.
- Figure 3a and c. Through the discussion in the text and the caption labeling, it is not clear if reactant 1a is present in the CVs or only water electrolysis is being observed because no organic substrate is in the electrolyte. This is more confusing because figure 3b and 3 are clearly with 1a as the reactant.
- Line 245. 'Conversion yields' is two separate terms.
- Line 256. Likely glassy carbon was used as the electrode.
- Line 251. The definition for yield is unconventional. Usually yield is the comparison of what desired product is obtained compared to how much reactant was added to the reactor. 'Theoretically formed tetrahydroquinoline product' appears to be only considering what had reacted of the initial quinoline, not all the quinoline that was added to the reactor.
- The wording, while understandable, is often awkward. Word choice and word order need to be polished throughout the manuscript.
- Supplemental information. Make sure all figures have captions that are descriptive of the experimental conditions, compounds studied, concentrations, electrolytes and duration of experiments.

A point-by-point response to the reviewers' comments

To reviewer 2:

Reviewer letter: This referee appreciates the authors' thoughtful responses. I think my previous concerns have been adequately addressed, so I recommend publishing this work in Nature Communications.

Answer: We highly appreciate the reviewer's positive comments on our manuscript.

To reviewer 3:

Reviewer letter: The revised manuscript by Bin Zhang, Cuibo Liu, and co-workers represents a hugely significant improvement on the original submission. Important previously unanswered questions have been addressed and discussed in a scholarly way with new supporting data. However, I remain concern on the methodology followed to quantify the conversion, yield and selectivity by GC. The equations used are correct, however, the problem arises from the fact that no internal standard was used. Authors based their calculations on standard calibration curves in which they represent peak area versus concentration. Although this method seems to be previously applied elsewhere, avoiding the use of internal (or external) standard is not a good practice of quantification in catalysis, especially when this is carried out in a GC-FID equipment. Actually, a correct calibration curve have to be constructed by representing: (molar ratio of substrate to the standard) versus (area ratio of substrate to the standard). Unfortunately, this means that authors have not properly worked, and consequently, all GC-based values reported on the manuscript should be revised before the manuscript being accepted, which, that to the reviewer's opinion, only should be occur if results do not differ too much from the previous ones. Furthermore, the following issues should also be addressed:

Answer: We do appreciate the reviewer's positive comments and constructive suggestions on our manuscript. According to the reviewer's suggestion, all GC-based values are now re-tested by adding dodecane as the internal standard, as often used in literature (e.g., *Nat. Catal.* 2019, 2, 71; *J. Am. Chem. Soc.* 2015, 137, 11718; *J. Am. Chem. Soc.* 2017, 139, 10790; *J. Am. Chem. Soc.* 2017, 139, 2035; *Angew. Chem. Int. Ed.* 2018, 57, 11262; *ACS Catal.* 2018, 8, 4545; *ACS Catal.* 2021, 11, 8197.). Taking the model substrate quinoline **1a** as an example, after three independent measurements, the average conversion determined by using correct calibration curves with dodecane as an internal standard is similar to that calculated by using standard calibration curves (98.2% vs. 99%). And, the average selectivity of product **2a** is calculated as 99% by using correct calibration curves with internal standard dodecane and 99% by using standard calibration curves, respectively. Both the errors of conversion and selectivity are less than

1%. Additionally, we have also compared the conversions of substrates **1l** and **1n** and selectivity of products **2l** and **2n** according to the correct calibration curves with an internal standard dodecane and the standard calibration curves, respectively. The conversion errors of **1l** and **1n** are 1.2% and 1.5%, and the selectivity errors of **2l** and **2n** are 0% and 0%, respectively. Thus, compared with the correct calibration curves with an internal standard dodecane, there only exist acceptable errors for the methodology of quantitative analysis using standard calibration curves in our previous version.

But, we agree with the reviewer's opinion that the internal standard is more reliable. Thus, to make our data more convincing, all the values of conversion and selectivity have been revised by using dodecane as an internal standard in the revised manuscript and Supplementary Information (SI).

Fig. R1 The standard calibration curves for quantitative analysis of **a** quinoline (**1a**) and **b** 1,2,3,4-tetrahydroquinoline (**2a**).

Fig. R2 The correct calibration curves with an internal standard dodecane for quantitative analysis of **a** quinoline (**1a**) and **b** 1,2,3,4-tetrahydroquinoline (**2a**).

Fig. R3 The standard calibration curves for quantitative analysis of **a** quinoxaline (**1**) and **b** 1,2,3,4-tetrahydroquinoxaline (**2**).

Fig. R4 The correct calibration curves with an internal standard dodecane for quantitative analysis of **a** quinoxaline (**1**) and **b** 1,2,3,4-tetrahydroquinoxaline (**2**).

Fig. R5 The standard calibration curves for quantitative analysis of **a** isoquinoline (**1n**) and **b** 1,2,3,4-tetrahydroisoquinoline (**2n**).

Fig. R6 The correct calibration curves with an internal standard dodecane for quantitative analysis of **a** isoquinoline (**1n**) and **b** 1,2,3,4-tetrahydroisoquinoline (**2n**).

Concerning other specific concerns or comments, we have provided a point-by-point response. To save the reviewer's valuable time, key revisions are displayed on a yellow background in the revised manuscript and SI. We are sure that the quality of this work will be greatly improved after being revised by considering these kind suggestions from the reviewers.

Comment 1: Comments on products 2a, 2b, 2c, 2d, 2f should be included in the manuscript.

Answer: Thanks for the reviewer's kind comments. We have added the corresponding comments on products 2a, 2b, 2c, 2d, and 2f to the revised manuscript. The descriptions are extracted and shown as

follows: “Quinoline and a variety of functionalized quinolines bearing electron-donating and electron-withdrawing groups on the benzene rings or pyridine rings can be transformed to the corresponding hydrogenated products with good to excellent conversions and moderated to high isolated yields (**2a-j**). To be specific, quinolines featuring methyl, methoxy, and amide groups in the 6-position all work well to deliver the corresponding THQ products (**2b-d**) in 85-90% isolated yields, and 6-fluoro-1,2,3,4-tetrahydroquinoline (**2f**) is also successfully prepared with 83% conversion.”.

Comment 2: The chemical structure of quinoline-6-carbonitrile and 1,2,3,4-tetrahydroquinoline-6-carbonitrile are wrongly drawn in Supplementary Figure 19.

Answer: We acknowledge the reviewer for pointing out this error. The chemical structures of quinoline-6-carbonitrile and 1,2,3,4-tetrahydroquinoline-6-carbonitrile have been revised in Supplementary Fig. 20. The revised version can be extracted and display below (Fig. R7).

Fig. R7 A qualitative analysis of possible products by GC-MS for electrocatalytic hydrogenation of quinoline-6-carbonitrile substrate.

Comment 3: Authors state in the response to referees' letter that “The amide group is compatible under our reaction conditions, giving rise to the corresponding product with 80% isolated yield.” However, a 90% isolated yield can be found in Table 1 of the manuscript.

Answer: Thanks a lot for the reviewer to point out this error. The amide group is highly compatible under our reaction conditions to give the 1,2,3,4-tetrahydroquinoline-6-carboxamide product in a 90% isolated yield. We have also carefully checked the manuscript and SI.

Comment 4: The meaning of the revised sentence “Furthermore, the phenyl ring is prone to reduction before initiating desired reaction sequences under electro- or photocatalytic reaction conditions” is different from the sentence of the former manuscript “Furthermore, phenyl ring is more prone to lose rather than to obtain an electron before initiating desired reaction sequences under electro- or photo-catalytic reaction conditions.”. Please, revise it!!

Answer: We acknowledge the reviewer’s kind comments. Compared with quinoline, the electron density of 1,2,3,4-tetrahydroquinoline is raised due to the electron-donating property of saturated N-heterocycle. Thus, the benzene ring in 1,2,3,4-tetrahydroquinoline is more likely to lose rather than obtain an electron before initiating subsequent reactions. It is harder to be further hydrogenated under our electrochemical conditions, improving the regioselectivity of electrochemical hydrogenation of quinoline.

We are sorry that our descriptions confuse the reviewer. We have revised the previous sentence to make it clearer in the revised manuscript. The descriptions are extracted and shown as follows: “Furthermore, due to the electron-donating property of saturated N-heterocycle in THQ, the benzene ring is more likely to lose rather than obtain an electron before initiating subsequent reactions under electro- or photo-catalytic reaction conditions.⁴³ It is harder to be further hydrogenated under our electroreduction conditions, thus improving the regioselectivity of electrocatalytic hydrogenation of quinoline.”.

We highly appreciate the reviewer’s thorough reading and comments/suggestions about our manuscript.

To reviewer 4:

Reviewer letter: There are still significant work that would need to be done before this manuscript could be considered for publication.

Answer: Thanks very much for the reviewer's comments on our manuscript. As for the concerns or comments of the reviewer, we have provided a point-by-point response. To save the reviewer's valuable time, key revisions are displayed on a yellow background in the revised manuscript and Supplementary Information (SI). We are sure that the quality of this work will be greatly improved after being revised.

Comment 1. It is still not clear to the reviewer why the work of the manuscript is so significant that it be worthy of Nature Communications instead of an organic chemistry or electrochemistry-based journal.

Answer: We appreciate the reviewer's comment. We are sorry to cause the reviewer's hesitation on the significance of our work at this time. However, the reviewer has given a positive comment on our work "The catalyst proposed in the work holds significant promise for its selectivity and activity. The work could be enhanced through how it is presented and changing at what point experiments are compared to each other.", "this manuscript have value based on higher selectivity of chemical's transformation" in the first round review. There might be some misunderstandings due to the imperfection of the comparative experiment and the inaccurate expression, leading to the negative comments of the reviewer. Here, we would like to explain it as follows:

First, we want to re-emphasize the brief background and significance of our work. Catalytic hydrogenation of quinolines has been a long-standing subject in synthesis chemistry, which provides a straightforward and convenient route to produce a variety of fine chemicals and drug-related skeletons. At present, transition metals catalyzed hydrogenation by using H₂ as the hydrogen source at given temperatures is still a prevailing method for selective hydrogenation of quinolines. Despite impressive achievements, the H₂ is mainly produced by the methane steam reforming technology, which is high energy consumption and accompanied by massive CO₂ release, and the safe storage and cost-effective transportation of H₂ remain a matter of concern. Additionally, hydrogenation of the benzene ring often occurs under thermocatalytic conditions, leading to the generation of 5,6,7,8-tetrahydroquinoline and decahydroquinoline byproducts. Therefore, searching for a **decarbonized and efficient method** for selective hydrogenation of quinolines by applying a **green and easy-to-handle hydrogen donor (e.g., H₂O) and a cost-effective catalyst at room temperature (RT)** is significant. Such a technique will achieve distributed manufacturing of hydrogenation of quinolines where H₂ as a hydrogen source is not readily available, thus complementing the state-of-the-art thermocatalytic hydrogenation of quinolines by using H₂.

Electrochemical hydrogenation by combining renewably sourced electricity and clean water is becoming a potential approach to obtain desired products because of its mild, efficient, and environmentally benign properties. Current studies are dominantly focused on the hydrogenation of CO_2 , NO_3^- , and other easily reducible organic substrates (e.g., nitros, aldehydes, ketones, alkynes, halides, and nitriles). However, the **electrochemical hydrogenation of quinolines with high resonance stability has been rarely touched and still faces a great challenge**. This may be ascribed to the lack of an efficient cathode material to activate quinolines.

Fortunately, we can circumvent the above issues by designing surface fluorine modified cobalt (Co-F) cathode via in situ electroreduction of $\text{Co}(\text{OH})\text{F}$ nanowire precursor to effectively activate the quinoline and H_2O , thus enabling highly selective hydrogenation of quinolines to 1,2,3,4-tetrahydroquinolines (THQs) at room temperature (RT). The significant points of our manuscript are listed below.

(1) An easily accessed, highly efficient, and durable fluorine modified cobalt (Co-F) cathode for electrocatalytic hydrogenation of quinolines to THQs.

- ✓ Co-F was designedly synthesized by the electroreduction of $\text{Co}(\text{OH})\text{F}$ nanowire precursors. A series of ex and in situ techniques characterized its formation process and structure.
- ✓ Co-F enabled electrocatalytic hydrogenation of quinolines to THQs with up to 99% selectivity and 94% isolated yield at RT with H_2O , outperforming the dominant Pt, Pd, and other cathodes.
- ✓ Co-F could be reused for 7 cycles with no obvious decay in the conversion of **1a** and selectivity of **2a**, revealing its stable performance.

(2) Combining theoretical and experimental data to unveil the high-performance origin.

- ✓ Theoretical calculations revealed that the enhanced adsorption of quinolines and promoted generation of active hydrogen atom (H^*) were two key factors for the hydrogenation process.
- ✓ $\text{F}^- \text{K}^+(\text{H}_2\text{O})_7$ networks were proposed to be formed to facilitate the formation of H^* , and K^+ and a high pH value were also very important for the good performance in this reaction.
- ✓ A more favorable flat adsorption of quinoline, and an unique 1,4/2,3-addition path involving H^* were proposed by combining control experiments and EPR test.

(3) Outstanding methodology universality and promising utility.

- ✓ 17 examples of quinoline and its derivatives were selectively hydrogenated with good to excellent isolated yields.
- ✓ Expedient synthesis of four deuterated *N*-heterocycles by using D_2O , and gram-scale production of two bioactive precursors demonstrated the practical utility.

- ✓ Paired synthesis of THQ and adiponitrile in a divided two-electrode electrolyzer at a lower voltage could be achieved, which was otherwise difficult to access by current approaches.

In addition, our work not only provides a mild and efficient approach for selective hydrogenation of quinolines but also offers a paradigm for designing highly active metal electrocatalysts with surface adsorbates to improve the reaction activity and product selectivity for other organic electrocatalytic transformations. Moreover, the fluorine effect in our work has been confirmed by performing other organic reactions, and the corresponding results have been added to the revised manuscript and SI.

Second, we do not think that this manuscript is more suitable for an organic chemistry or electrochemistry-based journal. The novelty of this work is to develop an efficient and sustainable method for selective hydrogenation of quinolines to THQs by applying green and abundant H₂O as the hydrogen donor over a cost-effective catalyst at RT, complementing the state-of-the-art thermocatalytic hydrogenation of quinolines by using H₂. Our work involves material design and preparation, organic synthesis, and water electrolysis, which is an important advance in nanochemistry, electrocatalysis, and organic synthesis. "**Nature Communications**" is a multidisciplinary journal and it covers the natural sciences, including physics, *chemistry*, earth sciences, medicine, biology, and all related areas. Two works on hydrogenation of quinolines in thermocatalysis separately catalyzed by a homogeneous Cobalt (Pang et al. *Nat. Commun.* 2020, 11, 1249.) and a single Iron catalyst (Long et al. *Nat. Commun.* 2020, 11, 4074.) have been recently published in "**Nature Communications**". We have also recently reported a work "*Converting copper sulfide to copper with surface sulfur for electrocatalytic alkyne semihydrogenation with water*, *Nat. Commun.* 2021, 12, 3881." in "**Nature Communications**", which is deemed as one of the cutting edge research published in the field of 'Catalysis' and highlight by the editor (<https://www.nature.com/collections/ihbfhbiibg>). This work on electrocatalytic hydrogenation of quinolines using H₂O represents an important breakthrough in electrosynthesis due to the hard activation of quinolines with high resonance stability.

Third, it should be pointed out that traditional thermocatalytic hydrogenation by using H₂ as the hydrogen source at high temperature is still a prevailing method for selective hydrogenation of quinolines to THQs. While electrocatalytic hydrogenation powered by renewable energy has gradually become an attractive and potential near-zero-emission approach to obtain desired hydrogenated products by utilization of safe and abundant water as a hydrogen donor under ambient conditions. In my opinion, thermocatalytic and electrocatalytic hydrogenation does not have a competing interest conflict and should go hand in hand. Since the electrocatalytic process owns many advantages under ambient conditions, which will complement the thermocatalytic hydrogenation using H₂, thus promoting the development of catalytic and synthetic chemistry. Importantly, because of the important applications of deuterated compounds, electrocatalytic deuteration will be more economic by using safe, low-cost, and easy-to-handle D₂O as the deuterated source than that of thermocatalytic deuteration by employing

unrecoverable and expensive D₂.

We believe that our manuscript is the original research of unusual urgency and significance in catalysis that appeals to a broad, general audience in the multidisciplinary fields of electrocatalysis, water electrolysis, materials, organic synthesis, and deuterated drug synthesis. Thus, **Reviewers 2 and 3** gave very positive comments on the novelty and importance of our work and provided professional revision suggestions on our work. Both of them recommended the publication of our work in '*Nature Communications*', the reviewer 2 said "*Given the importance and novelty, I recommend this work to publish in Nature Communications once the following issues are adequately addressed.*" The reviewer 3 mentioned that "*Based on the novelty of the synthetic strategy, the "a priori" impressive catalytic results, and the conceptual advance presented in this work, it might be suitable for Nature Communications.*".

Moreover, according to the reviewer's suggestion, we have re-written the "Introduction" part, provided new data on the recycling test of Co-F within 3 h using conversion as a function of time for each cycle, revised the corresponding descriptions to make them clearer, and sent our manuscript to the American Journal Experts (AJE) and asked a native English speaker to edit it. Thus, our revised manuscript is a greatly improved version, which will be better to meet the criteria of "*Nature Communications*".

Comment 2. Throughout the manuscript H₂ and water electrolysis are mentioned as positives for the work. In electrochemical hydrogenation, hydrogen generation is seen as a loss of electrons to undesired reactions, not a favorable occurrence. It is true that water activation is necessary for both electrochemical hydrogenation and water electrolysis. It is undesired to do the full reduction to H₂, however. This needs to be clarified and corrected throughout the manuscript, abstract and captions.

Answer: We acknowledge the reviewer's kind comments and suggestions. Electrocatalytic hydrogenation reaction by using clean and abundant H₂O as the hydrogen source provides an efficient and sustainable route to access the hydrogenated products under ambient conditions. We do agree with the reviewer's comments "*In electrochemical hydrogenation, hydrogen generation is seen as a loss of electrons to undesired reactions, not a favorable occurrence.*" and "*It is undesired to do the full reduction to H₂*". The activation of H₂O to form the active hydrogen species ($\text{H}_2\text{O} + \text{e}^- + * \rightarrow \text{H}^* + \text{OH}^-$) is an important step for both water electrolysis and electrochemical hydrogenation. In situ generated active hydrogen atom (H*) is used as the hydrogen source in electrocatalytic hydrogenation reactions, which provides a promising alternative to producing hydrogenated products. Thus, promoting the activation of water to generate more H* and inhibiting the formation of H₂ are highly significant to achieve efficient electrocatalytic hydrogenation of quinoline. So, to be clearer and more accurate, we changed "*water*

electrolysis” to “water activation” or “water dissociation” in the revised manuscript. The related descriptions are extracted and displayed below.

1) In the abstract: “*F* is revealed to enhance the adsorption of quinolines and promotes water activation to produce active atomic hydrogen (H^*) by forming $F^-K^+(H_2O)_7$ networks.”

2) On Page 5: “This indicates that the Co-F cathode is favorable for generating H^* via H_2O dissociation, which will benefit the hydrogenation of **1a**.”

3) On Page 9: “These results suggest that F^- plays promotional roles in both water dissociation and **1a** hydrogenation, which may be due to the enhanced adsorption of **1a** and activation of H_2O by F^- (Figs. 1a-c).”

4) On Page 9: “When we apply 1.0 M tetramethylammonium hydroxide (TMAH) solution as the electrolyte, inferior performances of both water dissociation and **1a** Conv. are expressed.”

5) On Page 9: “Furthermore, changing KOH to NaOH also degrades the activities of water dissociation and hydrogenation of **1a**.”

6) On Page 9: “However, when using Co foil without a surface F^- modifier as the cathode, no significant differences in water dissociation and **1a** Conv. are observed after replacing KOH with TMAH or NaOH (Supplementary Fig. 15).”

7) On Page 10: “These results may illustrate that the key role of surface F^- is to promote H_2O activation via the interactions between F^- and hydrated K^+ .”

8) On Page 10: “We rationalize that the better performance for KOH may be attributed to the production of more H^* by accelerating the activation of H_2O (Supplementary Fig. 16a).”

Comment 3. The revised justification of the work states that the reason to do electrochemical hydrogenation rather than traditional, well-developed thermocatalytic methods is because H_2 gas safety concerns and high temperatures of 100-150 °C. This does not motivate the chemical industry who would find 100-150 °C as low temperature and frequently works with hydrogen gas safely. Motivating factors to consider are decarbonization through use of renewable electricity and no need of hydrogen produced from methane steam reforming (a lot of CO_2 produced in H_2 generation), or distributed manufacturing where hydrogen as a utility is not readily available.

Answer: We acknowledge the reviewer’s kind comments. According to the reviewer’s suggestions, we have carefully re-written the previous introduction to make it more acceptable to the researchers and avoid possible disputes. The related part of the revised introduction is extracted and displayed below. “1,2,3,4-Tetrahydroquinoline (THQ) skeletons are prevailing and important structural motifs that

commonly reside in bioactive entities and natural products.^{1,2} At present, hydrogenation of quinolines catalyzed by transition metals using molecular hydrogen (H_2) as the hydrogen source at given reaction temperatures is still a prevailing method to synthesize the 1,2,3,4-tetrahydroquinolines, which has been a long-standing subject in synthetic chemistry and also served as a model reaction to evaluate the performance of a newly developed catalyst or hydrogenation system.³⁻¹³ For example, Corma and co-workers made an important advance in realizing highly chemo- and regioselective hydrogenation of quinoline derivatives over a nanolayered Co-Mo-S catalyst at 110-150 °C under 12 bar of H_2 .¹⁴ The Beller group fabricated the first heterogeneous N-doped carbon modified iron-based catalysts for selective hydrogenation of quinolines and (iso)quinolones by using 40-50 bar of H_2 at 130-150 °C.¹⁵ Despite these impressive achievements, the storage and transportation of flammable H_2 often need special apparatuses, cautious operations, and additional manpower input. Additionally, hydrogenation of benzene ring usually occurs under thermocatalytic conditions, leading to the generation of 5,6,7,8-tetrahydroquinoline and decahydroquinoline byproducts. Therefore, searching for an efficient and sustainable method for selective hydrogenation of quinolines to THQs by applying a green and easy-to-handle hydrogen donor (e.g., H_2O) at room temperature (RT) is highly significant. Such a technique will achieve distributed manufacturing of quinoline hydrogenation where H_2 as a hydrogen source is not readily available, thus complementing the state-of-the-art thermocatalytic hydrogenation of quinolines by using H_2 .

Recently, electrochemistry has become increasingly significant in the synthesis field because of its mild, efficient, and environmentally benign properties.¹⁶⁻²² Electrocatalytic hydrogenation powered by renewable electricity has gradually been proved to be an attractive and promising approach to obtain value-added hydrogenated products by direct utilization of clean and safe H_2O as the hydrogen donor.²³⁻²⁵ Importantly, deuterated chemicals with improved biological or physicochemical properties compared with their hydrogenated analogues due to the kinetic isotope effect of deuterium (D) can be economically and expediently synthesized by employing inexpensive and safe D_2O .^{26,27} Current studies are dominantly focusing on the hydrogenation of CO_2 , nitrate²⁸⁻³¹ and other easily reducible organic substrates (e.g., nitros, aldehydes, ketones, alkynes, halides, and nitriles).²³⁻²⁷ However, electrocatalytic hydrogenation of quinolines with high resonance stability still faces a technological challenge. The Lei group reported a well-designed electrochemical arylation of electron-deficient arenes through reductive activation using a Pt plate cathode. However, the yields of coupling products of quinoline derivatives are low.³² We speculate that these inferior performances may be ascribed to the lack of suitable materials to effectively activate quinolines. Given the importance of THQs and related compounds, it is highly desirable to synthesize an advanced material to efficiently promote the activation of quinolines and water, which will be conducive to boosting the activity and selectivity of quinolines hydrogenation."

Comment 4. Lines 64-67. ‘...the phenyl ring is prone to reduction before initiating desired reaction sequences under electro- or photocatalytic reaction conditions. Therefore, electrochemical hydrogenation will avoid the hydrogenation of phenyl rings of quinolines and improve regioselectivity.’ These two sentences conflict with each other.

Answer: We thank the reviewer for this comment. Compared with quinoline, the electron density of 1,2,3,4-tetrahydroquinoline is raised due to the electron-donating property of saturated *N*-heterocycle. Thus, the benzene ring in 1,2,3,4-tetrahydroquinoline is more likely to lose rather than obtain an electron before initiating subsequent reactions. It is harder to be further hydrogenated under our electrochemical conditions, improving the regioselectivity of electrochemical hydrogenation of quinoline.

We are sorry that our descriptions confuse the reviewer. We have revised the previous sentence in the revised manuscript, which is extracted and shown as follows: “*Furthermore, due to the electron-donating property of saturated N-heterocycle in THQ, the benzene ring is more likely to lose rather than obtain an electron before initiating subsequent reactions under electro- or photo-catalytic reaction conditions.*⁴³ *It is harder to be further hydrogenated under our electroreduction conditions, thus improving the regioselectivity of electrocatalytic hydrogenation of quinoline.*”

Comment 5. Figure 2d and discussion around recyclability of the catalyst. The catalyst is likely decaying during each run, though only cycles 8 and 9, did the decay become significant enough to notice at 6 hours. Conversion as a function of time for each of the cycles would be more useful in really seeing if the catalyst is stable. Alternatively, plotting at 3 hours where the reaction is not complete on fresh catalyst will allow the differentiation of the catalyst as it ages. Right now, the results presented in figure 2d are misleading.

Answer: Thanks for the reviewer’s comments. The recycling stability of a heterogeneous catalyst is of great importance for its practical applications. We agree with the reviewer’s comment “*The catalyst is likely decaying during each run, though only cycles 8 and 9, did the decay become significant enough to notice at 6 hours.*” After considering the kind suggestions of the reviewer, we now test the recyclability of Co-F at -1.1 V vs. Hg/HgO within 3 h for eight runs. The results are plotted by conversion as a function of time for each cycle. As shown in Fig. R8, although the conversion of **1a** gradually decreases from the sixth cycle within the initial 0.5 h, it can still approach 83% within 3 h in the former five cycles. Additionally, the recycling experiments in our screened optimal 6 h reveal no apparent decline in the conversion of **1a** and selectivity of **2a** in 7 runs. These results illustrate that although there is a slight decrease in the activity of the catalysts during each run at the initial hours, no obvious decrease in the conversion of **1a** is observed during the optimal time of 6 h according to the time-dependent **1a**

conversion experiments, demonstrating a relatively stable performance of Co-F. We have added the data and descriptions of recyclability tests of Co-F within 3 h to the revised manuscript. The revised descriptions of the recyclability of Co-F are extracted and displayed below.

*“Moreover, to evaluate the recyclability of Co-F, this electrode is repeatedly used for the next electrochemical experiments by adding the same amount of **1a** after being washed several times with ethanol and deionized water. Time-dependent transformations reveal that the conversion of **1a** gradually decreases from the sixth cycle within the initial 0.5 h (Fig. 2d). However, it can still approach 83% conversion within 3 h in the former five cycles. Additionally, we have also tested the performance stability of Co-F in our screened optimal 6 h (Supplementary Fig. 12). No apparent decline in the conversion of **1a** and selectivity of **2a** in 7 runs. These results may demonstrate a relatively stable performance of Co-F for electrocatalytic hydrogenation of **1a**. However, for the eighth and ninth runs, the decreased conversion of **1a** may be mainly ascribed to the loss of surface F (Supplementary Fig. 13) and the deactivation of Co catalytic sites caused by the nitrogen moiety of the substrate or product.”*

Fig. R8 Cycle-dependent conversions of **1a** over Co-F at -1.1 V vs. Hg/HgO within 3 h.

Additionally, the evaluation of recyclability in our work was carried out according to many important works published in top-tier journals (e.g., *J. Am. Chem. Soc.* 2020, 142, 962; *J. Am. Chem. Soc.* 2017, 139, 9419; *J. Am. Chem. Soc.* 2017, 139, 10790; *J. Am. Chem. Soc.* 2017, 139, 2035; *Angew. Chem. Int. Ed.* 2016, 55, 15656–15661; *Energy Environ. Sci.* 2020, 13, 4990–4999; *ACS Catal.* 2021, 11, 8197; *ACS Catal.* 2016, 6, 5816; *Chem. Sci.* 2018, 9, 8134.). In which they usually performed the recycling tests under their screened optimal reaction conditions to achieve nearly full conversion and high selectivity for several cycles. According to the obtained results (e.g., conversion and selectivity), the column with

nearly full conversion in each cycle is used to evaluate the stability of a heterogeneous catalyst. **So the discussion of recyclability of the catalyst in our work is reasonable and reliable.** Furthermore, the novelty of this work is to develop an efficient and sustainable method for selective hydrogenation of quinolines to THQs by applying green and abundant H₂O as the hydrogen donor over a cost-effective catalyst at RT, complementing the state-of-the-art thermocatalytic hydrogenation of quinolines by using H₂ rather than just investigating the stability of Co-F during the electrochemical process. The significance of this work can be found in reply to **comment 1** of the fourth reviewer.

Comment 6. Line 165. Catalyst is stated as being quickly used after synthesis. Is this indicating the catalyst is unstable?

Answer: Thanks for the reviewer's comments. Electroreduction of precursors offers an important addition to the toolbox of chemical conversion synthesis of highly active materials. The Pourbaix diagram (also known as the potential-pH diagram) of metals can guide the conversion of precursors to a certain phase of the reduced materials under the certain constant potentials and pH values of electrolytes. For the electroreduction of metal oxides, hydroxides, sulfides, or halides to metal, the precursor experiences a reconstruction with the leaching of oxygen/sulfur/halogen to generate the reduced metal with abundant low-coordination sites, which is highly active for catalysis. In our work, the Co-F cathode was synthesized through a facile electroreduction of Co(OH)F nanowire precursors at -1.2 V vs. Hg/HgO in 1.0 M KOH electrolyte, and the metallic Co phase with abundant low-coordination sites was real catalytic phase for electrocatalytic hydrogenation of quinoline. Therefore, to keep the high activity of Co-F and avoid its oxidation causing a decrease in activity, Co-F should be quickly used as the cathode for electrocatalytic hydrogenation of quinoline after its formation. Furthermore, the Pourbaix diagram also indicates that electrode material is stable under a certain range of potential and pH. Therefore, the Co-F can keep stable during the electrocatalytic hydrogenation of quinoline in this manuscript.

In brief, the as-prepared Co-F accessed via electroreduction of Co(OH)F is highly active. It is usually unstable when subjecting it out of certain conditions for its formation, such as exposure to the air, but it can keep stable under our electrochemical conditions according to the Pourbaix diagram.

Comment 7. Line 177. The catalysts in 2c are not acknowledged, only those in the SI Fig 8c are acknowledged.

Answer: Thanks for the reviewer's comments. We have added descriptions related to the Co-based catalysts and also revised the statements on other catalysts in the revised manuscript. The revised

version is extracted and displayed here.

*“Additionally, our Co-F cathode demonstrates the highest conversion of **1a** among all the tested Co-based materials. And, the other electrodes, including Pt, Pd, and Ni₂P display high hydrogen evolution reaction (HER) activities, and Cu and glass carbon (GC) show poor activities for HER, but all of them are all inferior to Co-F for the hydrogenation of **1a** under the same reaction conditions (Fig. 2c).”*

Comment 8. Line 203. The time at the measurement is being compared, not just the conversion should be indicated.

Answer: Thanks for the reviewer’s constructive comments. In our work, the comparison of **1a** conversion over Co foil with additional NaF and our Co-F in Fig. 3b is evaluated under the same potential and within the same reaction time. To make it clearer, we have revised the previous description, which is extracted and displayed as follows: *“However, the Conv. of **1a** is still much lower over the Co foil cathode after extra addition of NaF than that of Co-F (67.1% vs. 98.2%) within the same reaction time at -1.2 V.”*

Comment 9. Figure 3a and c. Through the discussion in the text and the caption labeling, it is not clear if reactant **1a** is present in the CVs or only water electrolysis is being observed because no organic substrate is in the electrolyte. This is more confusing because figure 3b and 3d are clearly with **1a** as the reactant.

Answer: Thanks for the reviewer’s kind comments. We are sorry that the caption of Fig. 3 confuses the reviewer. In our work, LSV curves for water electrolysis in Figs. 3a and 3c were recorded only in the electrolyte without adding **1a**. To be clearer, we have revised the caption labeling with its specific contents in Fig. 3 in the revised manuscript. They are extracted and shown as follows: **“Fig. 3 The effects of fluorine and cations in the electrolyte on electrocatalytic hydrogenation of **1a**. a** LSV curves of Co foil recorded in 1.0 M KOH without **1a** at a scan rate of 10 mV s⁻¹, and **b **1a** Conv. and **2a** Sel. over Co foil with and without NaF. Reaction conditions: **1a** (0.1 mmol, 14.28 mmol L⁻¹), a mixed solution of 1.0 M KOH/dioxane (6:1 v/v, 7 mL), -1.2 V, 6 h, RT. c** LSV curves of Co-F recorded in 1.0 M MOH (M = TMA⁺, Na⁺, and K⁺) electrolyte without **1a** at a scan rate of 10 mV s⁻¹, and **d **1a** Conv. and **2a** Sel. over Co-F. Reaction conditions: **1a** (0.1 mmol, 14.28 mmol L⁻¹), a mixed solution of 1.0 M MOH (M = TMA⁺, Na⁺, and K⁺)/dioxane (6:1 v/v, 7 mL), -1.1 V, 6 h, RT.”**

Comment 10. Line 245. ‘Conversion yields’ is two separate terms.

Answer: Thanks for the reviewer’s kind comments. We now use “Conversion” to replace “Conversion

yields ” in the revised manuscript.

Comment 11. Line 256. Likely glassy carbon was used as the electrode.

Answer: We thank you for the reviewer’s comments. Due to good electrical conductivity, high chemical stability, and nearly inert property toward electrochemical transformations, the glassy carbon (GC) electrode is an ideal electrode to investigate the fundamental redox behavior of the substrate or electrocatalyst during electrochemical reactions (*Nature* 2016, 535, 406; *J. Am. Chem. Soc.* 2021, 143, 12460; *Angew. Chem. Int. Ed.* 2019, 58, 9155; *Angew. Chem. Int. Ed.* 2019, 58, 12014; *ACS Catal.* 2017, 7, 2730.). According to the oxidation or reduction peak or oxidation/reduction peaks from the Cyclic Voltammetry (CV) or Linear Sweep Voltammetry (LSV) curve, the behavior of the substrate or electrocatalyst that loses or gains an electron will be determined. Because the GC was inert to electrocatalytic hydrogenation of quinoline (Fig. 2c), we used it to investigate the reduction process during electroreduction of quinoline. The LSV curve in Fig. 4b demonstrates a distinct peak (red line) when quinoline **1a** was added into the system, revealing its reduction by accepting an electron. Therefore, we may deduce that electrocatalytic hydrogenation of quinoline starts from an electron transferring to quinoline, supporting our proposed mechanism.

Comment 12. Line 251. The definition for yield is unconventional. Usually yield is the comparison of what desired product is obtained compared to how much reactant was added to the reactor. ‘Theoretically formed tetrahydroquinoline product’ appears to be only considering what had reacted of the initial quinoline, not all the quinoline that was added to the reactor.

Answer: Thanks for the reviewer’s comments. We are sorry that the descriptions in equation (3) confuse the reviewer. Here, the “*Theoretically formed tetrahydroquinoline product*” should be referred to “*Theoretically formed tetrahydroquinoline product after full consumption of quinoline*” in our manuscript. Actually, the amount (mol) of “*Theoretically formed tetrahydroquinoline product after full consumption of quinoline*” is the same as “*the added quinoline to the reactor*” as suggested by the reviewer. For better understanding, we have revised the previous equation (3) in the revised manuscript, which can be seen below.

$$\text{Yield (\%)} = \frac{n(\text{the obtained THQ product})}{n(\text{the added quinoline to the reactor})} \times 100\% \quad (3)$$

Comment 13. The wording, while understandable, is often awkward. Word choice and word order need

to be polished throughout the manuscript.

Answer: Thanks for the reviewer's kind comments. According to the reviewer's suggestion, we have sent our manuscript to the American Journal Experts (AJE) and asked a native English speaker to edit it. We believe that the presentation of our manuscript has been improved a lot. The corresponding letter of identification from the AEJ is displayed below.

✓ You are the author of this preprint
Surface Fluorine Promotes Electrochemical Transfer Hydrogenation of Quinolines with Water at Room Temperature

LANGUAGE QUALITY
Research Square's automated assessment helps you know where your manuscript stands.

Your current manuscript's score:

8 Above Average - Consider Final Improvements

- Your manuscript received a Language Quality Score of **8/10**.
- This manuscript has above-average English language quality compared to other academic papers we have evaluated. It may benefit from a revision to fix the remaining language errors.

Your edited manuscript's score:

9

- Our Digital Editing Tool made **210** changes to your document
- All edits were made using tracked changes that you can easily review and accept or reject
- Receive free digital edits for each subsequent version you submit to your journal

Download Your Edited File

Comment 14. Supplemental information. Make sure all figures have captions that are descriptive of the experimental conditions, compounds studied, concentrations, electrolytes and duration of experiments.

Answer: We acknowledge the reviewer's helpful suggestions. We have carefully checked and revised the SI and added the main reaction conditions, including compounds studied, concentrations, electrolytes, and duration of experiments to the figure captions of the revised SI.

We thank all kind and professional suggestions from the three reviewers. We are sure that the quality of this work will be greatly improved according to these nice comments and wise suggestions.

REVIEWER COMMENTS

Reviewer #3 (Remarks to the Author):

The reviewer's concerns have been satisfactorily addressed in the revised version of the manuscript. Therefore, I strongly recommend the acceptance of this work in Nature Communications.

Reviewer #4 (Remarks to the Author):

Comments: Revision significantly improves the manuscript.

In figure 2, it is not clear why part a is presented at 8 hours and most everything else in the document is 6 hours. The comparison to figure 2c and to other figures is not easily made because of the time differences.

Line 30. Be specific about temperature ranges rather than 'given reaction temperatures'

Line 238. This may be a case of overcorrection in the revision. 'Water dissociation' in this case is actually hydrogen evolution reaction, correct?

Line 422. Define NF here.

Some grammar still needs polishing. For example, "hard activation" is more commonly "difficult activation" and "designedly" I don't believe is a word. These are a few examples.

A point-by-point response to the reviewers' comments

To reviewer 3:

Reviewer letter: The reviewer's concerns have been satisfactorily addressed in the revised version of the manuscript. Therefore, I strongly recommend the acceptance of this work in Nature Communications.

Answer: We highly appreciate the reviewer's positive comments on our manuscript.

To reviewer 4:

Reviewer letter: Revision significantly improves the manuscript.

Answer: We thank the reviewer's positive comments on our manuscript. As for the concerns or comments of the reviewer 4, we have provided a point-by-point response. To save the reviewer's valuable time, key revisions are displayed on a yellow background in the revised manuscript and Supplementary Information (SI). We are sure that the quality of this work will be greatly improved after being revised.

Comment 1: In figure 2, it is not clear why part a is presented at 8 hours and most everything else in the document is 6 hours. The comparison to figure 2c and to other figures is not easily made because of the time differences.

Answer: Thanks for the reviewer's comments. We are sorry that Figure 2a in our manuscript causes a confusion to the reviewer. Fig. 2 displays how we screen the reaction conditions of electrocatalytic hydrogenation of quinoline with H₂O, including the optimal potential, reaction time, electrode comparison, and the recyclability test. In the beginning, we select a relatively long reaction time to screen the optimal potential. Fig. 2a reveals that the conversion of **1a** is low at -1.00 and -1.05 V vs. Hg/HgO within 8 h. Nearly full conversion of **1a** is obtained from -1.10 V vs. Hg/HgO, while there are a slight decrease in the conversion of **1a** at -1.25 and -1.30 V vs. Hg/HgO. Thus, -1.10 V vs. Hg/HgO is selected as the optimal potential.

To determine the exact reaction time, time-dependent experiments are further carried out at -1.10 V vs. Hg/HgO, which show that the nearly full conversion of **1a** can be finished within 6 h, shorter than the initially selected reaction time. This demonstrates that 6 h is enough for the electrocatalytic hydrogenation of **1a** at -1.10 V vs. Hg/HgO. Thus, 6 h is selected for sequent experiments.

In the revised manuscript, we revised the related descriptions. The key revisions are extracted and displayed below:

"Potential-dependent electrochemical experiments were carried out to screen the optimal potential for 8 h.", "This demonstrates that 6 h is enough for the electrocatalytic hydrogenation of **1a** at -1.10 V vs. Hg/HgO. Thus, 6 h is selected for sequent experiments."

Comment 2: Line 30. Be specific about temperature ranges rather than 'given reaction temperatures'.

Answer: Thanks for the reviewer's helpful suggestion. We have changed "given reaction temperatures" to "relatively high temperatures (usually ≥ 100 °C)" in the revised manuscript.

Comment 3: Line 238. This may be a case of overcorrection in the revision. ‘Water dissociation’ in this case is actually hydrogen evolution reaction, correct?

Answer: Thanks for the reviewer’s comments. Your comment is right. The “*Water dissociation*” is “*hydrogenation evolution reaction*” in original line 238. According to the kind suggestion, we revise this related description in the revised manuscript.

Comment 4: Line 422. Define NF here.

Answer: We acknowledge the reviewer’s kind comment. We have used “*Nickel Foam*” to define “*NF*” in the revised manuscript.

Comment 5: Some grammar still needs polishing. For example, “hard activation” is more commonly “difficult activation” and “designedly” I don’t believe is a word. These are a few examples.

Answer: We sincerely appreciate the reviewer’s wise comments and helpful suggestions. We have changed the “hard activation” to “difficult activation”, and deleted the inappropriate word “designedly” in the revised manuscript.

According to the reviewer and editor’s suggestion, we have sent our manuscript to the American Journal Experts (AJE) and asked a native English speaker to edit it. We believe that the presentation of our manuscript has been improved a lot. The corresponding letter of identification from the AEJ is displayed below. And the editing version of AJE is uploaded along with this revision.

We highly appreciate the reviewer's thorough reading and comments/suggestions about our manuscript.

We are sure that the quality of this work will be greatly improved according to these nice comments and wise suggestions.

REVIEWERS' COMMENTS

Reviewer #4 (Remarks to the Author):

The authors have adequately addressed the concerns previously brought up. Thank you.

A point-by-point response to the reviewers' comments

To reviewer 4:

Reviewer letter: The authors have adequately addressed the concerns previously brought up. Thank you.

Answer: We highly appreciate the reviewer's positive comments on our manuscript. We are sure that the quality of this work has been greatly improved according to these nice comments and wise suggestions. Thanks very much.